# HSP70s in Breast Cancer: Promoters of Tumorigenesis and Potential Targets/Tools for Therapy

**DOI:** 10.3390/cells10123446

**Published:** 2021-12-07

**Authors:** Alexander E. Kabakov, Vladimir L. Gabai

**Affiliations:** 1Department of Radiation Biochemistry, A. Tsyb Medical Radiological Research Center—Branch of the National Medical Research Radiological Center of the Ministry of Health of the Russian Federation, Koroleva 4, 249036 Obninsk, Russia; aekabakov@hotmail.com; 2CureLab Oncology Inc., Dedham, MA 02026, USA

**Keywords:** heat shock protein, glucose-regulated protein, mortalin, mammary gland, cancer stem cells, epithelial-to-mesenchymal transition, chemotherapy, radiotherapy, immunotherapy

## Abstract

The high frequency of breast cancer worldwide and the high mortality among women with this malignancy are a serious challenge for modern medicine. A deeper understanding of the mechanisms of carcinogenesis and emergence of metastatic, therapy-resistant breast cancers would help development of novel approaches to better treatment of this disease. The review is dedicated to the role of members of the heat shock protein 70 subfamily (HSP70s or HSPA), mainly inducible HSP70, glucose-regulated protein 78 (GRP78 or HSPA5) and GRP75 (HSPA9 or mortalin), in the development and pathogenesis of breast cancer. Various HSP70-mediated cellular mechanisms and pathways which contribute to the oncogenic transformation of mammary gland epithelium are reviewed, as well as their role in the development of human breast carcinomas with invasive, metastatic traits along with the resistance to host immunity and conventional therapeutics. Additionally, intracellular and cell surface HSP70s are considered as potential targets for therapy or sensitization of breast cancer. We also discuss a clinical implication of Hsp70s and approaches to targeting breast cancer with gene vectors or nanoparticles downregulating HSP70s, natural or synthetic (small molecule) inhibitors of HSP70s, HSP70-binding antibodies, HSP70-derived peptides, and HSP70-based vaccines.

## 1. Introduction

At present, breast cancer is the most commonly diagnosed female malignancy worldwide and one of the most frequent causes of cancer-related death among women (reviewed in [1,2]). Recent epidemiological data show that after the lung cancer, breast cancer is the second most common malignancy overall. According to the GLOBOCAN online database, there were about 2.3 million newly diagnosed female breast cancer cases and 685,000 breast cancer deaths worldwide in 2020 [2]. Such sad statistics delineate breast cancer as one of the greatest threats to modern humanity, so many physicians, pharmacologists and researchers in clinics and laboratories worldwide currently aim their efforts at fighting this challenge.

There are generally accepted methods of classification and sub-typing of breast cancer. According to the primary histological picture, all the cases of breast cancer can be categorized into the two groups, namely in situ carcinomas and invasive carcinomas, which are then sub-classified depending on their intragland localization and features including the tumor cell morphology, proliferative activity and grade of differentiation [3]. Breast cancer subtypes can be classified on the basis of molecular markers expressed by the tumor cells; such classification has a prognostic significance and is important for the choice of therapeutic strategy [3,4]. Immunohistochemical determination of a marker of cell proliferation Ki-67, estrogen receptor (ER), progesterone receptor (PR) and epidermal growth factor receptor 2 (ErbB2 or HER2/neu) allows oncologists to distinguish five major subtypes of breast cancer: (1) luminal A (low Ki-67 level, ER and/or PR positive, and HER2 negative); (2) luminal B (high Ki-67 level, ER and/or PR positive, and HER2 positive); (3) basal-like or triple negative (ER, PR and HER2 negative); (4) HER2-enriched (ER and PR negative, and HER2 positive); (5) normal-like (similar to luminal A subtype: low Ki-67 level, ER and/or PR positive, and HER2 negative) [3,4]. In the cases of triple-negative breast cancer, such tumors are more aggressive (quickly growing, invasive and metastatic), insusceptible to hormonal and targeted therapy, highly resistant to chemotherapy and imply poor prognosis. In contrast, luminal A and B, normal-like and HER2-enriched subtypes are treatable with tamoxifen or aromatase inhibitors for ER+/PR+ patients and trastuzumab or lapatinib for HER2/neu patients and have more favorable prognosis. Luminal A and normal-like subtypes are similar slowly growing tumors but prognosis of normal-like breast cancer is slightly worse than that of luminal A [3,4]. 

Several cell lines have been characterized which, being derived from the different breast cancer subtype specimens, enable modeling of either subtype in in vitro studies and as the tumor xenografts growing in nude mice: MCF-7 and T47D cells corresponding to luminal A (ER and PR positive, HER2 negative), BT474 cells corresponding to luminal B (ER positive, HER2 positive), MDA-MB-231, MDA-MB-468, and BT549 cells corresponding to triple negative, etc. [5]. 

Over the past decade, largely thanks to the thorough surveillance aimed at early diagnosis of breast cancer and advances in the tumor treatment, patient outcomes and prognoses of breast cancer have been somewhat improved [1,2]. Nevertheless, breast cancer still remains an extremely acute problem of modern medicine. Besides the particularly serious cases of triple negative breast cancer, other subtypes of breast carcinomas may be complicated by the early metastasis spread in the lymph nodes and bones and also the tumor cell resistance to chemotherapy and radiotherapy [1,2,3,4].

As well as in other types of human malignancies, the so-called cancer stem cells (CSCs) play the key role in the development/course of breast cancer [4,6,7]. This fraction of the tumor-initiating cells is thought to be responsible for the constant renewal of breast cancer cell population and unlimited malignant growth with invasion and metastasis formation; additionally, CSCs exhibit high chemoresistance and radioresistance along with their ability to evade immune attacks [6,7]. Breast CSC-like cells, which are a product of the so-called epithelial-to-mesenchymal transition (EMT) under the influence of intratumoral hypoxia and other microenvironmental and humoral factors (and/or some therapeutics used), also aggravate breast cancer pathogenesis by driving the invasion into adjacent organs, metastasis formation and therapy resistance [8,9]. 

Obviously, the development of new approaches to more successful treatment of breast cancer including its advanced and metastatic forms is of paramount importance. In this respect, it seems fruitful to identify and exploit some of breast cancer-promoting proteins as molecular targets which would help to better treat this malignancy. Among such breast cancer-promoting proteins, there are the members of heat shock protein 70 subfamily (HSP70s or HSPA) which seem especially critical and potentially targetable. 

Thus far, there are two publications on transcriptomic and proteomic studies that demonstrate a diversity of the in vivo HSPA expression in different subtypes of human breast cancer (see [10,11] and Table 1). 

Apparently, HSP1A and HSPA8 are expressed in all breast cancer subtypes; HSPA2 is higher in luminal, while HSPA5 and HSPA6 are higher in basal subtypes. In turn, ER+/PR+ status is associated with HSP1A, and HER2+ status—with HSPA8 (Table 1). A significance of these associations for development/pathology of breast cancer subtypes is yet unclear. 

This review summarizes and analyzes data regarding multifaceted implication of HSP70s in the initiation and pathogenesis of breast cancer. Interplay between HSP70s and oncogene products, HSP70-mediated alterations in the regulation of cell cycle, apoptosis and signaling, HSP70-promoted stimulation of tumor growth and metastases, HSP70-conferred breast tumor resistance to the immune system and therapeutics, and other relevant points are reviewed herein. Additionally, some intracellular and cell surface-exposed HSP70s are considered as druggable or antibody-accessible targets for treatment of breast cancer. Potential methods and various agents for curative targeting/use of HSP70s in the fight against breast cancer are discussed. 

## 2. Members of HSP70 Subfamily: General Characteristics, Localization, Functioning

The members of 68–78 kDa HSP subfamily (DnaK or HSPA, or HSP70s) are the major ATP-dependent chaperones of eukaryotes. The molecular structure of all HSP70s is rather conservative and exhibits the common domain organization with the N-terminal nucleotide-binding domain (NBD) and C-terminal substrate-binding domain (SBD) connected to each other by a flexible linker (Figure 1). In turn, the SBD is divided on a peptide-binding pocket and a bendable lid (see Figure 1 and [12,13]); these two subdomains allow the chaperone to transiently clasp a substrate protein molecule. 

As for the subcellular localization of different species of HSP70s, these ubiquitous chaperones can reside and function in the cytoplasm, nucleus/nucleoli, endoplasmic reticulum (ER) and mitochondria, at ribosomes, cytoskeletal frameworks, proteasomes and lysosomal membranes, and also on the cell surface and in the intercellular space including the cargo of cell-secreted exosomes. This section provides a brief description of the major features of mammalian HSP70s and regulation of their expression/functioning. 

### 2.1. Heat Shock Cognate Protein 70 (HSC70) and Inducible HSP70

The so-called heat shock cognate protein 70 (HSC70 or HSPA8) is the constitutively expressed form of HSP70s which is largely present in the cytoplasm. Under non-stressful conditions, cytosolic HSC70 catalyzes polypeptide chain folding in an ATP-dependent manner under maturation, transport or degradation of protein molecules. Mechanistically, the ATPase reaction with the binding and hydrolysis of ATP in the NBD followed by release of ADP and inorganic phosphate ensures the energy charge for HSP70-mediated (re)folding of protein substrates. In vivo, the interactions of HSC70 with ATP, ADP and protein substrates are regulated by (co-)chaperones and co-factors such as HSP40, HSP90, Hip, Hop, CHIP, BAG-1, BAG-3 and others that catalyze either folding and stabilization or degradation of various client proteins in the course of cyclic work of the ATP-consuming chaperone machine [13]. 

Additionally, the functions of HSC70 and its inducible forms can be regulated at the level of post-translational modifications of the chaperone molecule (phosphorylation, acetylation, methylation, ubiquitination, and others) [14]. One more method of the regulation of activities of HSP70 is its dimerization and oligomerization, since a dynamic pool of monomers, dimers, and oligomers of this HSP can exist in the cytosol [15]. All the above three ways of regulation of HSP70, namely at the level of (i) its interactions with co-chaperones and co-factors, (ii) its post-translational modifications and (iii) its oligomerization imply a possibility of multitarget inhibiting functions of HSP70 with different cell-permeable agents (see further Section 5). 

In the unstressed mammalian cell, the typical substrates of mammalian HSC70 are nascent polypeptides growing at ribosomes, clathrin and heat shock transcription factor 1 (HSF1). Similar activities, responses and subcellular localizations are also attributed to another member of the HSPA subfamily such as HSPA14 (HSP70L1). There are data that suggest the acquisition of radioresistance by human breast cancer cells is accompanied by the enhancement of HSPA14 expression [16]; however, a causal link between the intratumoral HSPA14 level and tumor radiation response has not been yet established. Together with HSP90, HSC70 binds to and inhibits HSF1, thus preventing the HSF1 activation under normal conditions. When denatured and aggregated proteins accumulate within the stressed cell, they recruit HSC70 and HSP90 from their complexes with HSF1 and the latter is activated to trigger the heat stress transcriptional response with subsequent expression of all stress-inducible *HSP* genes [17,18]. 

In the case of a proteotoxic stress, the inducible forms of HSP70 (often referred to as HSP72 and encoded by three very closely related paralogs: HSPA1A, HSPA1B, and HSPA1L) are expressed as a result of the stimulation of HSF1-mediated stress response that confers the improved survival/recovery and stress tolerance in HSP-enriched cells [13]. HSPA2 (HSP70-2) is known to be involved in spermatogenesis and also plays a role in breast cancer (considered in detail in Section 4.1). One more stress-inducible form of HSP70 is HSPA6 whose cellular functions are less explored [19]. It is thought that the HSP70-conferred protection of stressed cells is mainly due to the ability of excess HSP70 (i) to attenuate the proteotoxicity by chaperoning stress-damaged cellular proteins and their aggregates [13] and (ii) to prevent the stress-induced cell death by blocking pathways that lead to apoptosis or necroptosis [20,21]. Besides its direct (inhibitory) interactions with the effectors of apoptosis [20,21], HSP70 can stabilize the inhibitor apoptosis proteins (IAPs) such as c-IAP1 and X-linked IAP (XIAP) [22]. In addition to its cytoprotective chaperoning of cell death regulators, HSP70 is able to modulate cellular signaling and protein degradation thereby maintaining the viability and fitness of stressed cells including the cancerous ones [20,21,23]. 

The inducible forms of HSP70 (HSPA1A, HSPA1B, and HSPA1L) are actively implicated in carcinogenesis as well as pathogenesis of cancer; many human malignancies exhibit the enhanced HSP70 expression which is commonly correlated with their aggressiveness and resistance to therapeutics [12,20,21] (see further Section 3). A role of HSPA6 in cancer remains to be clarified. HSP70-enriched cancer cells are better adapted to the stressful conditions of the tumor microenvironment such as hypoxia, nutrient limitation, acidosis, etc. [24]. Notably, HSP70 can be exposed at the surface of malignant cells and also be secreted or incorporated into the extracellular vesicles (exosomes) [12,21,25]. A certain contribution of HSP70 to the development and pathogenesis of breast cancer is considered in the next sections.

### 2.2. Glucose-Regulated Protein 78 (GRP78) 

The 78 kDa glucose-regulated protein (GRP78) (sometimes referred to as HSPA5 or binding immunoglobulin protein (BiP)) is an ATP-dependent chaperone and a member of the HSP70 subfamily; its molecule domain organization is similar to that of HSC70/HSP70 (see Figure 1). In vivo, GRP78 is implicated in the mechanisms of folding, degradation, transport and secretion of (glyco)proteins and also in the control over gene transcription, signaling pathways, Ca^2+^ homeostasis, autophagy and apoptosis [26]. 

GRP78 residing in the ER is known to be a cellular sensor of ER stress and master regulator of the so-called ‘unfolded protein response’ (UPR) that is triggered under hypoxia, hypoglycemia, Ca^2+^ ion imbalance, inhibition of protein glycosylation and other conditions inducing ER stress. In the unstressed cell, the constitutively expressed form of GRP78 within the ER lumen is in the complexes with the ER-specific stress signal transducers (SSTs) which are inactive being in that bound state. Under ER stress, GRP78 is recruited from those complexes by unfolded proteins and thus releases the three SSTs: protein kinase RNA-like endoplasmic reticulum kinase (PERK), activating transcription factor 6 (ATF6) and inositol-requiring enzyme 1 (IRE1); then the liberated SSTs are activated and trigger the UPR that leads to the stimulation of certain signaling pathways and expression of ER stress-responsive genes (reviewed in [24,26]). Among protein products of the GRP78-controlled UPR there are calreticulin, components of the ER-associated protein degradation (ERAD) machinery serving for the proteolysis and also inducible GRPs (GRP170, GRP94, GRP78, GRP75) which in an ATP-dependent fashion catalyze either the disaggregation/refolding or splitting of stress-damaged proteins within the ER and mitochondria. The intracellular level of GRP78 can be downregulated via the ubiquitination of GRP78 followed by its degradation, which is regulated by the acetylation/deacetylation of the chaperone molecule [27]. A minor part of GRP78 can be translocated to the cell surface or secreted outside the cell (including GRP78 being a component of exosomes), whereas a cytosolic isoform of GRP78 (GRP78va) can be produced via alternative splicing [24,26,28].

Intriguingly, GRP78 may sometimes promote cell death: if some GRP78-mediated responses to ER stress are cytoprotective (induction of chaperones, chaperone-mediated refolding or degradation of stress-damaged proteins, prevention of apoptosis), the other ones are proapoptotic (expression of CHOP and activation of caspases) [26]. Such ambivalent outcomes of the GRP78-controlled UPR appear to ensure the post-stress fulfilment of ‘quality control’ program: saving/recovery of stressed cells with reparable damages and apoptotic elimination of severely injured cells. 

Both intracellular and extracellular GRP78 are involved in the mechanisms of malignant growth, cancer stemness maintenance, EMT, invasion and metastasis spread, and tumor resistance to therapeutics [26,28,29]. GRP78 and the GRP78-controlled UPR drive the metabolic reprogramming of cancer cells residing in the hypoxic regions of tumors [24,30]. Roles of GRP78 in breast cancer and its potential as of a targetable protein to combat this disease are reviewed in the next sections. 

### 2.3. Glucose-Regulated Protein 75 (GRP75 or Mortalin)

GRP75, also named mortalin or HSPA9, or mitochondrial HSP75 (mtHSP75), or peptide-binding protein 74 (PBP74), is largely localized to the mitochondrial compartment where this ATP-dependent chaperone ensures the import of nuclear gene products. In cooperation with HSP60, GRP75 participates in the intramitochondrial protein folding and assembling of multimolecular protein complexes, thus maintaining the functional activity of mitochondria [28]. Moreover, mammalian GRP75 is thought to carry out the control of intracellular Ca^2+^ homeostasis, cell proliferation and senescence; by means of its direct interactions with p53 onco-suppressor protein, GRP75 regulates such p53-mediated events as the cell cycle arrest, DNA break repair and apoptosis or senescence following genotoxic stresses [29]. 

Importantly, the overexpression of GRP75 (including HSPA9B or mortalin-2) is found in many human malignancies which is causally correlated with their invasive growth, metastasis spread and resistance to chemotherapy and radiotherapy [24,29,31,32] (see also Section 4.3). Tumoral GRP75 is known to contribute to the cancer stemness and EMT program that are associated with the aggressiveness and persistence of mammary gland carcinomas [29,33,34]. Additionally, GRP75 (mortalin) may present on the cancer cell surface, plays roles in cellular membrane trafficking and is a typical protein component of exosomes secreted by tumors of different origin including the breast tumors [35]. It is described in the next sections how GRP75 is implicated in the breast cancer pathogenesis and how this chaperone may be targeted to fight breast cancer. 

## 3. Role of HSP70 in Oncogenesis and Tumor Progression

Numerous observations demonstrate that HSP70 is often overexpressed in a wide range of human cancers, including breast cancer, and this correlates with tumor malignancy progression and worse survival (see below). Furthermore, knockdown/knockout of HSP70 genetically or pharmacologically is much more devastating to cancer cells than to normal cells (see ref [36] for review). All these data lead to an idea that HSP70 may be intimately involved in tumor development and progression.

Thus far, there has been a study on Rat-1 fibroblasts indicating that HSP70 can be an oncogene per se [37]. However, this phenomenon was not reported in other cell lines and apparently is just a peculiarity of these cells which are more prone to transformation. Thus, HSP70 does not appear to be an oncogene by itself. By analogy with ‘oncogene addiction’, there is also a phenomenon called ‘non-oncogene addiction’ when some genes, not being oncogenic by themselves, are required for tumorigenesis [38]. The HSP70-encoding gene seems to be one of such genes—by itself, it does not lead to tumorigenesis but is absolutely required for it. Initially, it was suggested that tumor cells need chaperones because they undergo permanent proteotoxic stress [39]. Indeed, the microenvironmental conditions in some tumor regions are rather harsh due to acidic pH and low levels of oxygen and nutrients. This can explain why tumor cells require more chaperones for survival, and the HSP levels increase when cancer progresses [24,39,40]. However, a noxious microenvironment cannot explain why tumor cells still need HSP70 in vitro when, as was shown recently, proteotoxic stress either varies markedly between breast cancer cell lines [41] or is undetectable [42]. Furthermore, HSP70 is required not only for maintaining the tumor cell viability, but for the malignant transformation itself. 

Tumor-initiating CSCs, whose occurrence is thought to be a result of oncogenic mutations in normal stem cells or progenitors, appear to need the high HSP70 level in order to constantly maintain their ‘stemness’, thus contributing to tumor development/persistence (reviewed in [29]). 

### 3.1. HSP70 and Cell Transformation

There are several intrinsic mechanisms which are considered to protect cells from oncogene-induced transformation and therefore are necessary to prevent tumor formation. Accordingly, if HSP70 can disrupt those mechanisms, this will promote the cell transformation and tumorigenesis. 

#### 3.1.1. HSP70-Mediated Protection from Oncogene-Induced Apoptosis 

The protective function of HSP70 against apoptosis caused by variety of stresses is well-established (see refs [21,43] for review). Initially, it was suggested that a main function of HSP70 in cell transformation is to protect from oncogene-induced apoptosis. Indeed, expression of some oncogenes such as c-myc can sensitize cells to apoptosis, and HSP70 overexpression can protect from apoptosis (e.g., induced by cytotoxic drugs). For example, in myc-overexpressing Rat-1 and U937 cells, HSP70 is protective against etoposide and campothecin [44]. In a recent study in osteosarcoma cells, c-myc-induced apoptosis was blocked by sBAG-1, a small isoform of BAG-1, a co-chaperone of HSP70 which was required for anti-apoptotic protection [45]. In mice, c-myc was shown to cause mammary gland tumorigenesis associated with increased Bax-dependent apoptosis [46] but so far a role of HSP70 in protection from apoptosis during mammary gland transformation has not been studied. The suggested involvement of HSP70 in promoting the myc-driven tumorigenesis is shown in Figure 2. 

Interestingly, mice overexpressing HSP70 live shorter lives than wild-type mice due to the development of lung tumors, and this was associated with decrease in caspase 9 and increase in Bcl-2 expression, pro-apoptotic and anti-apoptotic proteins, respectively [47]. 

#### 3.1.2. HSP70-Mediated Protection from Oncogene-Induced Senescence

The most common effect of oncogene expression in cells is senescence rather than apoptosis. Oncogene-induced senescence is also regarded as a protective mechanism against oncogenesis [48,49]. Although senescent cells do not die as apoptotic ones do, they undergo complete growth arrest and thus cannot propagate to form tumors. Many anticancer drugs and radiation in low doses cause senescence rather than apoptosis [50,51,52], and HSP70 can protect from it [53,54]. Furthermore, oncogenes which are often overexpressed in breast cancer such as HER2, PIK3CA, or RAS also cause senescence rather than apoptosis [55,56,57,58]. These finding led to hypothesis that the main function of HSP70 in cell transformation can be prevention of oncogene-induced senescence. 

Indeed, while HSP70 knockdown led to senescence in breast cancer cell lines expressing PIK3CA, HER2 or RAS oncogenes, normal human breast epithelial cell line MCF10A was resistant to HSP70 knockdown. However, when MCF10A cells were transformed with PIK3CA, RAS, HER2 oncogenes, they undergo senescence/growth arrest [59]. There are at least three mechanisms by which HSP70 prevents oncogene-induced senescence. In wild (wt) p53 cells, senescence was associated with activation of p53/p21 pathway, whereas in mutant (mut) p53 cells it was associated with activation of extracellular signal-regulated kinase (ERK); also, senescent cells demonstrate loss of survivin independent of p53 status. HSP70 can prevent activation of these pathways by oncogenes [59], thus promoting the oncogenic cell transformation (see Figure 2). 

Importantly, the phenomenon of HSP70-mediated protection from oncogene-induced senescence was demonstrated not only in vitro, but in a mouse model as well. Meng et al. [60] established a mouse model in which HSP70 knockout (KO) is combined with expression of HER2. In these experiments, they used mice with both orthologs of human HSP70, HSP70A (HSPA1A) and HSP70B (HSPA1B), deleted and termed HSP70−/− [61]. Those animals were bred with mice-expressing Her2/neu (neu—a rodent homolog of HER2) under the control of mouse mammary tumor virus (MMTV) promoter (MMTVneu) [62]. Thus, WT neu+/−, HSP70+/−neu+/−, and HSP70−/−neu+/− mice were generated, and tumor development in these animals was followed. There was similar tumor incidence between heterozygous HSP70+/−neu+/− and WT-neu+/− mice (median tumor appearance in this strain was 66 weeks), indicating that one copy of the HSP70 genes is sufficient to support mammary tumorigenesis induced by HER2. In contrast, the absence of HSP70 in the homozygous KO animals almost completely blocked mammary tumor development: within >100 weeks no tumors appeared in 10 out of 11 animals, and in one HSP70−/−neu+/− mouse a tumor appeared by 88 weeks of age, which is far later than tumor appearance in control animals. Very importantly, strong β-galactosidase staining indicative of extensive senescence was observed in mammary ducts from HER2-expressing HSP70/KO animals. Senescence was so strong that it led not only to suppression of hyperplasia, but also to suppression of normal alveolar branching [60]. Therefore, this animal model confirms that HSP70 is critical for HER2-induced tumor development, and this is associated with suppression of senescence.

In another study, using the same HSP70 KO mouse strain, Gong et al. [63] investigated a role of HSP70 in mammary carcinogenesis induced by another oncogene, polyomavirus middle T (PyMT) oncogene under control of MMTV promoter. This oncogene is much more powerful, and mice rapidly develop tumors which metastasize in lungs. Similar to the HER2 mouse model, HSP70 heterozygotes develop tumors similarly to wild type [63]. Complete knockout of HSP70, although it did not completely prevent tumor development as in the HER2 model, significantly decreased their size, especially at early time points (25–40 days after onset of the oncogene expression) [63]. The effect of HSP70 knockout on tumorigenesis in this model was not as drastic as in the HER2 model probably because PyMT oncogene disables oncogene-induced suppressors such as p53 and, hence, requirement for HSP70 decreases. However, the effect of HSP70 knockout on metastases in this model was much more drastic (see further Section 3.2.3). 

#### 3.1.3. HSP70 and Tumor Evasion of Immune System 

Evasion of the immune system is a critical part of oncogenesis: the healthy immune system usually can recognize transformed cells and eliminate them. As the result of this, cancer incidence is in humans relatively low despite formation of millions of potentially transformed cells every day. To evade recognition by immune cells and their attack, transformed cells employ several mechanisms. Among them is decreased MHCI expression and presentation of cancer-associated antigens, disabling of cytotoxic T lymphocytes (e.g., via suppression of the immune system by expression of immune checkpoint inhibitors such as PD1), and acquiring resistance to cell death evoked by immune cells. HSP70 appears to either increase or have no effect on MHCI expression [64,65,66]. Evidently, HSP70 by itself is a very immunogenic protein, and it is often used in design of, for example, various vaccines as an immune adjuvant (see further). Furthermore, the extracellular (plasma membrane) expression of HSP70 by cancer cells makes them more vulnerable to immune attack by cytotoxic T-lymphocytes [66], natural killer (NK) cells [67,68] and macrophages [69,70]. 

While cell surface HSP70 stimulates the immune system, membrane-associated HSP70 from tumor-derived exosomes (TDE) appears to mediate immune-suppressive functions of myeloid-derived suppressor cells (MDSC); this effect is due to IL-6-dependent STAT3 activation [71]. Furthermore, suppression of TDE formation by amiloride markedly enhanced antitumor effect of cyclophosphamide in mouse mammary adenocarcinoma TS/A and other mouse tumors. Such HSP70-expressing exosomes were also found in urine of breast cancer patients [72]. Similar to amiloride, inhibition of HSP70 with A8 peptide aptamer increased the antitumor effect of cisplatin and 5-FU via the activation of the immune system [72]. 

Unlike cell surface HSP70, intracellular HSP70 appears to protect tumor cells from immune attack [73,74]. In vitro, HSP70 was shown to save tumor cells from tumor necrosis factor (TNF) and other members of TNF family such as TRAIL and FASL which are secreted by macrophages and other immune cells; this protection was associated with suppression of both apoptotic and non-apoptotic cell death [75,76,77,78]. In in vivo rodent cancer models, depletion of HSP70 by antisense treatment increased sensitivity of tumor cells to macrophage-mediated toxicity and suppressed their growth in animals; such sensitization appears to be associated with an increase in macrophage-generated NO/ROS [79,80,81,82]. Accordingly, in a model with syngeneic colon tumor MC38, a novel inhibitor of HSP70 AP-4-139B (see further Section 5.1) activated CD8/CTL infiltration, apparently via secretion of DAMP by inhibitor-treated tumor cells [83]. Administration of ADD70, a peptide inhibitor of HSP70, in syngeneic rat colon tumors PROb also activated CD8 cells and suppressed their growth, while in the nude (i.e., immune compromised) animals their growth was unaffected [84]. In another syngeneic mouse model, an HSP70 inhibitor, peptide P17, caused regression of B16 melanoma in the animals which was associated with infiltration of tumors with immune cells, i.e., T-cells, NK cells, dendritic cells and macrophages [85]. Therefore, intracellular HSP70 and exosomal HSP70 are able to protect some tumors from the host immune system, but the HSP70 expression on the tumor cell surface can stimulate the anti-tumor immune response. Probably, a similar interplay between HSP70 and immune system takes place in the case of breast cancer. 

### 3.2. HSP70 and Tumor Progression

#### 3.2.1. HSP70-Mediated Promotion of Angiogenesis 

To grow and invade surrounding tissues, cancer cells should induce formation of new blood vessels (neovascularization) to obtain nutrients and oxygen. One of the main regulators involved in this process is hypoxia-inducible factor 1 (HIF1) [24]. Colvin et al. [86] demonstrated that HSP70 is indispensable for activation of HIF1 in human breast cancer MCF-7 cells since knockdown of HSP70 completely blocked HIF1 induction in response to hypoxia mimetics. As the result, generation of vascular endothelial growth factor (VEGF), a major factor promoting angiogenesis, was markedly suppressed [86]. The effect of HSP70 on HIF1 induction appears to be mediated by HuR and HSP70 co-chaperone BAG3 [86]. The other effect of HSP70 on HIF1 is protection of the latter, along with HSP90, from von Hippel-Lindau protein (pVHL)-independent degradation [87,88]. 

Besides VEGF, interleukin-5 (IL-5) was found to be another angiogenic activator. HSP70 knockdown in vitro led to dysfunction of IL-5-induced proliferation, migration, colony tube formation of human umbilical vein endothelial cells (HUVEC), a commonly used in vitro model of angiogenesis. Furthermore, IL-5-mediated angiogenic response was blocked in HSP70-knockout mice [89]. 

Whereas endogenous HSP70 appears to promote neovascularization via HIF1/VEGF and IL-5 generation, extracellular HSP70 can also stimulate angiogenesis. It promotes HUVEC migration and tube formation in vitro, and microvessel formation in vivo similarly to VEGF [90]. This effect of HSP70 is mediated via its interaction with CLEC14a (C-type lectin domain family 14 member) [91]. 

Thus, HSP70 is involved in angiogenesis by at least three mechanisms: activation of HIF1/VEGF by tumor cells, and stimulation of angiogenesis in stromal endothelial cells either directly or via secretion of IL-5. 

#### 3.2.2. HSP70-Mediated Promotion of Epithelial-to-Mesenchymal Transition (EMT), Migration, and Invasion 

The deadliest hallmark of tumor progression is metastases, and it is metastases in vital organs rather than primary tumors that kill about 90% of patients. For instance, in the case of breast cancer, DCIS (ductal carcinoma in situ) cannot metastasize and it is the least malignant form of breast cancer, whereas triple-negative breast cancer easily metastasizes and is regarded as the most lethal [1,2,3]. The formation of metastases is a very complicated process which takes years to occur and includes, among others, the resistance to anoikis, EMT, cell migration, invasion into the vasculature, and an ability of circulating cancer cells to anchor and grow in remote tissues. 

Knockdown of HSP70 was shown to increase anoikis of tumor cells placed in poly-hema coated plates implicating HSP70 in anoikis protection [92]. Extracellular expression of HSP70, while making tumor cells sensitive to immune attack (see above), can promote EMT in some models [93], although intracellular HSP70 can inhibit it [92]. 

Extracellular HSP70 can also stimulate cell migration and invasion via several mechanisms. One of them involves tissue transglutaminase (tTG) which play an essential role in cell migration. During migration of triple negative human breast cancer MDA-MB-231 cells, tTG is redistributed to leading edges of the cells and this depends on HSP70 [94]. Another mechanism of the HSP70-mediated invasion consists of assisting HSP90-dependent activation of matrix metalloproteinase-2 (MMP-2) [95], an enzyme involved in cell migration, invasion and metastases. When HSP70 was depleted or inhibited, the MMP-2 activity in MDA-MB-231 cells and their invasion were markedly suppressed [95]. Similarly, HSP70 was shown to assist in surface membrane translocation of alpha-enolase, an enzyme shown to contribute to the motility and invasiveness of cancer cells through the protein non-enzymatic function of binding plasminogen and enhancing plasmin formation [96]. An interesting study has been done recently by Nigro et al. [96]: the authors treated human breast cancer cells with recombinant HSP70 from *Arabidopsis tiliana* (r-AtHSP70). Since HSP70 from plants shows strong similarity to that from humans, it may confer a similar effect. Indeed, extracellular r-AtHSP70 activated MMP-9 in MB231 cells. It also increased the motility and invasion of MCF-7 and MDA-MB-231 human breast cancer cells, but not MCF10A cells from the normal human mammary gland epithelium [97]. 

Besides extracellular functions, intracellular HSP70 is also involved in cell invasion. For instance, HSP70 along with HSP90 is essential for stabilization and activation of the WASF3 metastasis-promoting protein. In particular, the motility of MDA-MB-231 cells was impaired by HSP70 knockdown, HSP70 inhibitors, or knockdown of WASF3 [98]. Another client of HSP70 essential for cell migration and invasion is FAK, since HSP70 inhibitors impaired FAK-mediated tumor cell migration and invasiveness [99]. In mouse breast cancer cells, HSP70 knockout significantly decreased their migration and invasion [63]. 

#### 3.2.3. HSP70-Mediated Promotion of Metastases

The question arises whether these in vitro effects of HSP70 is related to in vivo phenomenon of metastasis either in mouse cancer models or in clinics. There are several studies addressing this question. Using a xenograft model of human breast cancer, Kluger et al. [100] assessed differentially expressing proteins in a GI101A parental clone and its highly metastatic variant GILM2. HSP70 was found to be significantly downregulated in the metastatic variants rather than upregulated [100]. Although HSP70 showed lower expression in GILM2 cells compared with GI101A cells in the gene array results, the tissue microarrays of breast cancer patients showed a strong association between high HSP70 expression and tumor aggression, i.e., lymph node metastases (*p* = 0.0002) and survival (*p* = 0.05) [100]. Similarly, in recent study, HSP70 was significantly higher in women with metastatic breast cancer than in those with non-metastatic breast cancer (*p* = 0.001) [101]. 

Sun et al. [102] established metastatic variants of syngeneic spontaneous breast cancer in TA2 mice. They found that expression of HSP70 was upregulated in metastases of mouse breast cancer. Furthermore, in human breast cancer samples, HSP70 expression was closely associated with breast cancer metastasis: for instance, in triple-negative breast cancer samples, the expression of HSP70 in breast cancer that developed metastasis was significantly higher than that in breast cancer that did not metastasize [102]. 

Using in vitro assays and mouse xenografts, Lee et al. [103] showed that aggressive properties of triple negative breast cancer cell lines can be induced by TNFα by via upregulation of A20 (TNFAIP3) protein. In a striking contrast, TNFα induces a potent cytotoxic cell death in luminal (ER+) breast cancer cell lines which fail to upregulate A20 expression in response to TNF. Importantly, overexpression of A20 not only protects luminal breast cancer cell lines from TNFα-induced cell death via induction of HSP70-mediated anti-apoptotic pathway but also promotes a robust EMT phenotype and metastases by activating the pSTAT3-mediated inflammatory signaling. Accordingly, treatment with HSP70 inhibitor VER155008 increased the TNF-induced cell death and suppressed pro-inflammatory signaling [103]. 

Being correlative, these reports do not directly address the question whether modulation of HSP70 levels affect metastatic ability of tumors in vivo. This was clarified by experiments of Gong et al. [63] with HSP70 knockout mice. In this work PyMT oncogene caused rapid developments of breast tumors which were partially delayed by HSP70 knockout (Section 3.1.2 above). At the same time, the effect of HSP70 knockout on lung metastases was much more drastic: at 110–120 days after tumor initiation, there were about 30 colonies (metastases) in wt mice, but no colonies formed in HSP70 knockout mice; in heterozygotes, metastasization was similar to that in wt. Such impaired tumorigenesis and metastasis in HSP70-knockout mice was associated with downregulation of the Met gene and reduced activation of the oncogenic c-Met protein [63]. 

Therefore, all the data coming from in vitro studies, mouse models and clinics demonstrate that, besides its involvement in the malignant transformation, HSP70 also promotes tumor progression and metastases. 

#### 3.2.4. HSP70 in Tumor Stroma

It is now obvious that surrounding tumor tissues play an important role in the pathogenesis of cancer: depending on stroma, tumor growth can be either promoted or suppressed. On one hand, tumor cells can modify stroma (e.g., via secretion of angiogenic factors such as VEGF, see above); on the other hand, stroma can affect tumor cells by either stimulating their growth (e.g., via tumor-associated fibroblasts and macrophages) or destroying them via the immune system. 

A recent study was demonstrated that, besides being necessary for growth of cancer cells in vitro and tumors in vivo, HSP70 expression in stromal cells is also essential for tumor growth [104]. Whereas mouse breast carcinoma E0771 efficiently forms tumors when transplanted into wt allogeneic mice, there was no tumor formation in HSP70 knockout mice. This effect of HSP70 knockout may be caused suppression of infiltration of macrophages in tumors. Furthermore, macrophages from the HSP70 knockout animals have lower migration in vitro compared to those from wt animals. Given that HSP70 affects migration of tumor cells (see above), this effect of HSP70 knockout on macrophages is not surprising. Interestingly, even if tumor cells in vitro were resistant to HSP70/BAG-3 inhibitor JG-98 (see further Section 5.1), they became sensitive to it in vivo due to the decreased infiltration of macrophages [104]. Therefore, the inhibition of HSP70 in stroma is sufficient for antitumor effects even if cancer cells per se are resistant to such inhibition. 

## 4. Role of other HSP70 Subfamily Members in Tumorigenesis

### 4.1. HSPA2 (HSP70-2)

HSPA2 is more than 80% homologous to HSP70 (HSPA1) and was initially considered as a testis-specific chaperone crucial for male germ cell development and function [105,106]. The latter data also implicated it in cancer, in particular breast cancer [10,11,107].

Rohde et al. [108] were the first to find higher expression of HSP70-2 in human breast cancer cell lines and tissues, which was later confirmed by Scieglinska et al. [107]. Depletion of HSP70-2 caused growth arrest and senescent-like morphology in breast tumor cells, while normal breast epithelial MCF10A cells were resistant to such depletion [108]. Furthermore, knockdown of HSP70-2 appears to upregulate the p53/p21 pathway, similar to what was seen with HSP70 depletion (see Section 3.1 above). Additionally, HSP70-2 depletion in human breast cancer MCF-7 cells caused downregulation of LEDFG (lens epithelium-derived growth factor) and lysosomal cell death associated with this effect, whereas normal MCF10A were resistant to it. Interestingly, LEDFG expression, similar to HSP70-2, was much higher in human breast cancer comparing to normal tissue [109]. 

In a recent systematic study, HSP70-2 expression was detected in a majority of breast cancer patients (83%) irrespective of various histotypes, stages and grades [110]. HSP70-2 expression was also observed in breast cancer cell lines with different receptor status: BT-474 and SK-BR-3 (HER2-positive), MCF-7 (ER-positive), and MDA-MB-231 (triple negative breast cancer), but not human normal mammary epithelial HNMEC cells [110]. Similar to HSP70 knockdown, HSP70-2 knockdown led to growth arrest, senescence, apoptosis, loss of colony-forming ability and inhibited growth of MDA-MB-231 xenografts. Furthermore, HSP70-2 was shown to be essential for cancer cell motility, migration and invasion, which also resembles the respective activities of HSP70 [110]. 

Of note, however, and contrary to above cited studies, is that Soika et al. [111] did not observe any effects of knockdown of HSP70-2 on cell growth, migration, adhesion, and invasion of MCF-7 cells. Yet, a recent study of Yang et al. [112] supported the essential role of HSP70-2 in tumorigenesis. They found that RING finger protein 144A (RNF144A), a member of the RING-in-between-RING family of E3 ubiquitin ligases, functions as a tumor suppressor in breast cancer and target HSPA2 for degradation. RNF144A was downregulated in a subset of primary breast tumors and restoration of RNF144A suppressed breast cancer cell proliferation, colony formation, migration, invasion in vitro, tumor growth, and lung metastasis in vivo. In contrast, knockdown of RNF144A promoted malignant phenotypes of breast cancer cells. Notably, the ligase activity-defective mutants of RNF144A impaired its ability to induce ubiquitination and degradation of HSPA2, and to suppress breast cancer malignant phenotype as compared with its wild-type counterpart. Moreover, RNF144A-mediated suppression of breast cancer cell proliferation, migration, and invasion was rescued by ectopic HSPA2 expression. Clinically, low RNF144A and high HSPA2 expression in breast cancer patients was correlated with aggressive clinico–pathological characteristics and decreased overall and disease-free survival [112]. 

Therefore, HSPA2 seems to be involved in the pathogenesis of breast cancer by sustaining viability, promoting motility, migration and invasion. Apparently, although HSP70 is a close homologue of HSPA2, it cannot completely substitute its function, and vice versa, and therefore knockdown of either HSPA2 or HSP70 has profound effect on survival and tumorigenicity of breast cancer cells. 

### 4.2. HSPA5 (GRP78)

#### 4.2.1. Intracellular HSPA5

HSPA5 (GRP78 or BiP) is an important chaperone involved in folding/maturation of a bulk of proteins which are translated in the ER (see Section 2.2). It is induced under harsh conditions when protein folding in ER is impaired, e.g., during hypoxia or glucose deprivation, which is manifested as the UPR. Since such conditions happen quite often during growth of solid tumors [24], it is non-surprising to expect that tumors would induce HSPA5 during their growth. Indeed, several studies clearly demonstrated by immunohistochemistry and proteomic analyses that human breast tumors express much higher levels of HSPA5 than normal tissue [113,114,115]. What is unusual is that even when grown in vitro where no noxious factors are present, tumor cells continue to express high levels of HSPA5 [116]. Apparently, as in case with other chaperones described above, tumor cells do not tolerate HSPA5 depletion because of their addiction to it.

In contrast to inducible HSP70, whose knockout is well-tolerated, homozygous disruption of the GRP78 allele led to early embryonic lethality, while heterozygous GRP78 mice with half of wt GRP78 level are comparable to wt siblings in growth and development [117]. To determine the role of GRP78 in endogenous tumor growth, the GRP78+/−mice were crossed with the transgenic mice (MMTV-PyVT) expressing the polyoma middle T oncogene (PyMT or PyT) driven by the murine mammary tumor viral promoter [118], similar to those used by Gong et al. (see [63] and Section 3.1.2.), and these mice were monitored for tumor development. In GRP78+/+PyT mice, most tumors were first detectable between week 8 and 10, whereas most tumors in the GRP78+/−,PyT mice became detectable between week 10 and 12 [117]. Furthermore, tumor growth was significantly reduced in the GRP78+/−,PyT mice compared with the GRP78+/+,PyT mice. Of note, the size and morphology and weight of major organs in all four groups of mice (GRP78+/+,PyT, GRP78+/−,PyT, GRP78+/+, and GRP78+/−) were comparable, indicating that GRP78 heterozygosity primarily affects growth of tumors rather than normal tissues. Furthermore, tumors in situ from GRP78 heterozygotes showed enhanced apoptosis, and, when propagated in vitro, they grow slower than those from wt mice [117]. Thus, this study shows strong dependence of tumors on GRP78 when even loss of one copy of it impedes tumor development and growth [117]. This is in striking contrast to HSP70 where only homozygotes demonstrated this effect (see Section 3.1.2). 

Myb oncogene upregulation is associated with ER-positive breast cancer. In mouse mammary tumor virus (MMTV)-NEU mice, described above (Section 3.1.2.) where tumors are initiated by activation of HER2, MYB deletion was sufficient to abolish the tumor formation [119]. In the more aggressive MMTV-PyMT model system, MYB deletion delayed tumorigenesis significantly. Interestingly, GRP78 was induced by MYB [119], and, accordingly, GRP78 accumulation was observed in HER2-induced tumors [120], although its significance in these models was not further explored. The tumorigenesis-promoting interplay between PyMT, Myb, HER2 and GRP78 is schematically delineated in Figure 3. 

Ubiquitin-Specific Protease 22 (USP22) was identified as a member of the so-called “death-from-cancer” signature predicting therapy failure in cancer patients. In a recent study, Prokakis et al. [121] demonstrated that USP22 is required for the tumorigenic properties in murine and human HER2+-breast cancers. GRP78 was found that to be stabilized by USP22 and suppresses UPR-induced apoptosis in HER2-positive human breast cancer cell lines SKBr3 and HCC1954 [121] (see also Figure 3). 

BRCA1 is a well-known tumor suppressor which is mutated in a high percentage of familial breast cancer. Yeung et al. [122] reported that GRP78 is a downstream target of BRCA1: upregulation of wild-type BRCA1 suppressed the expression of GRP78, whereas expression of mutant BRCA1 gene or targeted inhibition of endogenous BRCA1 enhanced GRP78 expression (Figure 3). Furthermore, forced expression of GRP78 stimulated cell proliferation and prevented apoptosis induced by endoplasmic reticulum stress and chemotherapy in breast MCF-7 cancer cells while overexpression of wild-type BRCA1 could increase the apoptosis of GRP78-overexpressing cells. Conversely, knockdown of GRP78 by small interfering (si) RNA sensitized breast cancer cells to apoptosis, and this was reduced when the expression of BRCA1 was simultaneously knocked down by siRNA, indicating that BRCA1 also negatively regulates GRP78-mediated cell survival and resistance to apoptosis [122]. Therefore, the above data indicate that GRP78 is implicated in the breast cancer development associated with diverse oncogenes (see Figure 3). 

As with many other chaperones, GRP78 affects a number of processes involved in tumor occurrence and progression. Knockdown or inactivation of GRP78 was shown to suppress migration and invasion in diverse breast cancer cell lines [28,123,124,125]. Although not assessed directly in breast cancer, GRP78 was shown to promote hypoxia-induced EMT in A549 lung cancer cells [96]. In particular, when GRP78 was silenced, the levels of EMT markers fibronectin, vimentin, Snail 1/2, Twist 1/2 and ZEB2 were decreased and EMT was inhibited [126]. Accordingly, Nayak et al. [127] described that IKM5, an inhibitor of GRP78 (see also Section 5.2.1), inhibited cell invasion and EMT markers MMP-2, Twist1, and vimentin in MDA-MB-231 breast cancer cells. IKM5 was also shown to suppress lung metastasis of a 4T1 mouse breast carcinoma model [127]. Furthermore, the E1-adenovirus antigen-mediated suppression of metastatic activity of MDA-MB-231 cells was due to the deacetylation and degradation of HSPA5 [27,123,124]. When GRP78 was assessed in primary mouse tumors and matched distant metastasis, minimal GRP78 protein expression was observed in epithelial cancer cells of 67NR and 66cl4 mouse primary tumors which are weakly metastatic. In contrast, the high levels of GRP78 expression were detected in the aggressive 4T1.2 −primary tumor that spreads to several sites after primary tumor growth in the mammary gland and persisted in matched, spontaneous metastases within the lung, heart, kidney, and bones [128].

It was found that GRP78 is apparently required not only for tumor initiation, but also in tumor progression by regulating tumor angiogenesis. Dong et al. [129] were the first who found that tumor microvessel density (MVD) was significantly (70%) lower in GRP78 heterozygotes compared with wt mice in PyMT tumor model described above [117]. Further study revealed that, when wt syngeneic mammary tumor cells were injected into the host, GRP78+/−mice suppressed angiogenesis, tumor growth and metastasis during the early phase but not during the late phase of tumor growth [129]. Conditional heterozygous knockout of GRP78 in the host endothelial cells showed severe reduction in tumor angiogenesis and metastatic growth, with minimal effect on normal tissue MVD. Furthermore, knockdown of GRP78 expression in immortalized human endothelial cells showed that GRP78 is a critical mediator of angiogenesis by regulating the cell proliferation, survival, and migration [129]. Accordingly, GRP78 was shown to be upregulated in HUVEC in response to VEGF, and its knockdown suppressed the VEGF-induced cell proliferation, phosphorylation of extracellular-regulated kinase 1/2 (ERK1/2), phospholipase C-c, and VEGF receptor-2 [130]. 

Thus, similar to HSP70 (see Section 3.2.4 above), for tumor growth GRP78 is required to be expressed not only in tumor cells itself, but in tumor stroma as well. Additionally, like HSP70 (see Section 3.2.1), GRP78 seems to be one of the regulators of tumor-associated angiogenesis. 

#### 4.2.2. Cell Surface HSPA5

Besides its largely intracellular (ER) localization, GRP78 is often expressed on the surface of cancer cells and this expression seems to be also connected with cancer pathogenesis. The high levels of surface GRP78 (sGRP78) facilitated proliferation and migration, as well as suppressing apoptosis in human breast cancer MCF-7 cells [131]. Neutralization of sGRP78 by a specific antibody attenuated the sGRP78-induced cell growth and migration (see further Section 6.2.2). Importantly, high phosphorylation levels of the signal transducer and activator of transcription 3 (STAT3) were found in human breast tumors that that express sGRP78 and MCF-7 cells infected with adenovirus encoding human GRP78, while genetic and pharmacological inhibition of STAT3 reversed the impacts of GRP78 on cell proliferation, apoptosis, and migration [131]. These findings indicate that STAT3 mediates sGRP78-promoted breast cancer cell growth and migration. Furthermore, STAT3 activation was shown to be mediated by the CD44v transmembrane protein via its interaction with COOH-terminal proline-rich region (PRR) of GRP78 (see Figure 4). Prevention of such interaction caused decrease in cyclin D1 protein and increase in apoptosis [132]. 

Another partner of interaction with sGRP78 is such a protein as dermcidin (DCD), a cell-secreted growth and survival factor, whose gene was amplified and overexpressed in a subset of breast tumors [133,134]. Lager et al. [135] showed that GRP78 and DCD cooperate to regulate breast cancer cell migration that is dependent on the cell surface functions of these proteins (see Figure 4). Furthermore, Wnt/β-catenin signaling was identified as an important downstream intermediate in regulating this cell migration-defining phenotype [135]. 

Zhang et al. [136] have shown that MCF-7 breast cancer cells resistant to hormonal therapy overexpress sGRP78. Importantly, sGRP78 forms complex with phosphoinositide 3-kinase (PI3K), and the overexpression of sGRP78 promotes phosphoinositide 3-phosphate (PIP3) generation, indicative of PI3K activation. An insertion mutant of GRP78 at its N-terminus domain, while retaining stable expression and the ability to translocate to the cell surface as the wild-type protein, exhibited the reduced complex formation with p85 and production of PIP3 [136]. Such overexpression of sGRP78 and sGRP78-mediated regulation of the PI3K signaling, if occurring in vivo, may contribute to the mechanism of breast cancer resistance to hormonal therapy (see Figure 4). 

Finally, taking into consideration the great importance of CSCs and CSC-like cells for the development/pathogenesis of breast cancer [6,7,8,9], it is necessary to mention about a role of both GRP78 and sGRP78 for the cancer stemness (reviewed in [28,29]). Operating with MDA-MB-231 and MCF-7 cells, Conner et al. [28] have revealed that the GRP78 overexpression in the breast cancer cells is associated with an occurrence of CD24-/CD44+ tumor-initiating cells (TICs) and sGRP78+ breast cancer cells, exhibiting the expression of stemness genes, are a subset of TICs. It was also found that the sGRP78+ (CSC-like) breast cancer cells have an enhanced capacity to form multiple metastases in organs of mice [28]. 

#### 4.2.3. GRP78 and Breast Tumor Evasion of Host Immunity

Cook et al. [107] assessed the effect of GRP78 depletion by morpholino antisense oligonucleotides in LLC9 human breast carcinoma xenografts. They found that GRP78 knockdown decreased CD47 expression, a potent “do not eat me” signal, accompanied by increased macrophage infiltration [137]. Interestingly, unlike breast tumor tissue, inhibiting GRP78 in normal mammary tissue had an opposite effect, i.e., increased expression of the CD47. These data suggest a differential role of GRP78 in regulating CD47 signaling in neoplastic versus normal tissues [137]. 

Soto-Pontoja et al. [138] found that inhibition of the GRP78 in 4T1B breast cancer cells increased the cytolytic capacity of RAW264.7 macrophages targeting these cells. Conditioned media from GRP78-silenced breast cancer cells increased expression of tumor-attacking M1 macrophages (with CD80+ marker) and reduced the M2 marker, Arg-1, while conditioned media from control cells showed elevated macrophage CD206+ (an M2-like macrophage marker) [137]. Furthermore, GRP78 heterozygous tumor-bearing mice displayed increased CD68/CD80 co-localization when compared with wild-type tumors, suggesting increased infiltration of the tumors by M1-like macrophages. Thus, these data indicate that GRP78 reduction enhances macrophage attack and clearance of cancer cells [138].

Recently, Chen et al. [139] performed intravital imaging using a secreted GRP78 (secGRP78)-overexpressing mouse breast cancer cell line (E0771) and found that, when these cells secreted GRP78, it interacted with dendritic cells (DCs) and F4/80C macrophages in the liver. Importantly, overexpression of secGRP78 inhibited DC activation and induced M2-like (tumor-promoting) polarization in F4/80C macrophages. Moreover, secGRP78 overexpression enhanced production of TGF-β (a known immunosuppressor) in the liver. Thus, secGRP78 promotes tumor cell colonization in the liver by remodeling the tumor microenvironment and promoting immune tolerance of the tumor [139]. 

Finally, GRP78 was found to interact with immune checkpoint protein PD-L1 [140]. PD-L1, along with PD1, suppresses T cell-mediated immune responses, including the cytolytic activity of CD8 T cells which is critical for antitumor effect. Therefore, blockade of PD1/PD-L1 by antibody is currently widely used in cancer immunotherapy including therapy of triple negative breast cancer. In BT-549 breast tumor cells, GRP78 knockdown was shown to downregulate PD-L1 levels, while its overexpression upregulates PD-L1 levels apparently via promoting its stability. Furthermore, dual-high levels of GRP78 and PD-L1 expression were found to correlate with poor relapse-free survival in triple negative breast cancer [140]. 

Therefore, the above data demonstrated that both intracellular and extracellular expression of GRP78 can suppress both the innate and adaptive immunity which may explain why its high expression in breast cancer correlates with tumor progression and poor prognosis (see below). 

#### 4.2.4. Clinical Data 

In study of Zheng et al. [141] in breast cancer patients, GRP78 expression was negatively correlated with disease-free survival (*p* < 0.001). It is also positively correlated with progression (*p* < 0.05), metastasis (*p* < 0.005) and poor prognosis (*p* < 0.0001), and its expression in triple negative breast cancer is much higher than in other subtypes of breast cancer (*p* = 0.006) [141]. Accordingly, GRP78 expression was significantly associated with invasive, distant metastasis and proliferation of triple negative breast cancer (*p* < 0.05). In addition, the high expression of GRP78 was significantly associated with decreased disease-free survival (DFS) in patients with triple negative breast cancer (*p* < 0.001) [142]. In another study, GRP78 staining was also correlated with tumor grade (*p* < 0.0001), although no higher expression in triple negative breast cancer was found [143]. Interestingly, along with intracellular GRP78, sGRP78 was correlated with tumor grade (*p* = 0.047), stage (*p* = 0.001), and number of positive lymph nodes (*p* = 0.04) [131]. 

### 4.3. HSPA9 (GRP75 or Mortalin)

HSPA9 (GRP75), also widely known as mortalin, is a mitochondrial chaperone of the HSP70 subfamily (see Section 2.3). Wadhwa et al. [144] were the first to find that mortalin is overexpressed in different tumor cell lines, in particular human breast carcinomas MCF-7, MDA-MB-415, MDA-MB-436, MDA-MB-468, MDA-MB-361 and others; furthermore, overexpression of mortalin was observed also in human breast cancer tissues. When mortalin was artificially overexpressed in MCF-7 cells which have low tumorigenicity and cannot grow in nude mice, such mortalin overexpression enabled their growth. Mortalin-overexpressing MCF-7 cells also showed the enhanced migration in chemotaxis assay [144]; these data indicate that mortalin can contribute to the tumorigenicity.

Those data were later supported by other studies. Overexpression of mortalin in MDA-MB-231 cells increased their proliferation, migration and invasion; proteins known to play key roles in cell migration and EMT as well as proteins involved in focal adhesion, PI3K–Akt, and JAK–STAT signaling, were upregulated in the mortalin-expressing breast cancer cells [145]. Accordingly, expression levels of the mesenchymal markers vimentin, fibronectin, β-catenin (CTNNB1), CK14 (KRT14), and hnRNP-K were also increased upon mortalin overexpression, whereas the epithelial markers E-cadherin (CDH1), CK8 (KRT8), and CK18 (KRT18) were downregulated. Furthermore, the mortalin downregulation with GRP75-targeting small hairpin (sh) RNA or its inhibitor MKT-077 (see also Section 5.3) suppressed the migration/invasive capacity of breast cancer cells and was associated with a diminished EMT gene signature [145].

In a recent study of Zhang et al. [33], knockdown of mortalin in MCF-7 and SKBR3 breast cancer cells suppressed their migration, as well as expression of MMP-2 (which is involved in metastasization), and VEGF, a key factor of angiogenesis. When studying EMT process, the authors found that mortalin knockdown decreased expression of mesenchymal markers vimentin, Slug, and Twist while increasing expression of epithelial markers E-cadherin and ZO-1; apparently, this effect was mediated by the Wnt/β-cathenin signaling pathway [33]. In another publication from 2021 [34], knockdown of mortalin in MCF-7 and MDA-MB-231 cells was shown to significantly inhibit cell proliferation, migration and EMT; a spheroid-forming capacity and expression of stemness-associated genes also became suppressed in the mortalin-depleted breast cancer cells of both lines. Further experiments included an MCF-7 xenograft model and revealed that mortalin promotes EMT and maintains breast cancer stemness via activation of the Wnt/glycogen synthase kinase-3β (GSK3β)/β-catenin signaling pathway in vivo and in vitro [34]. The studies described in refs [33,34] are in consistent with each other and complement the above study [145]. Taken together, those findings [33,34,145] support a role for mortalin in breast cancer cell migration, invasion, and induction of EMT. As CSCs and EMT-generated CSCs-like cells are the pivotal players in the development/pathogenesis of breast cancer [6,7,8,9], the established involvement of mortalin in inducing EMT [33,34] and maintaining the stemness [34] in breast cancer cells seems to be of importance in terms of the search for appropriate targets to pharmacologically attack the malignancy (see Section 5.3). 

It is probable that, to activate the Wnt/β-catenin pathway [33,34], mortalin can involve other breast cancer-related mechanisms which promote EMT, invasion and metastases. In particular, the experiments with shRNA-induced knockdown of GRP75 (HSPA9) in triple negative breast cancer MDA-MB-231 cells revealed the HSPA9-dependent mechanism of haematological and neurological expressed 1-like (HN1L)-mediated upregulation of the expression of high-mobility group protein B1 (HMGB1) that was associated with the EMT program, tumor invasion and metastasis formation in a xenograft model [146]. 

The question arises how these experimental studies are related to clinical observations on breast cancer. Jin et al. [147] systematically studied expression of mortalin in invasive ductal carcinoma of breast. A total of 155 invasive ductal carcinomas of breast patients with strict follow-up, 52 ductal carcinomas in situ (DCIS) and 45 adjacent non-tumor breast tissues were selected for immunohistochemical staining of mortalin protein. Mortalin showed a mainly cytoplasmic staining pattern in breast cancers and the strongly positive rate of mortalin was in 64% of invasive ductal carcinoma of breast and was significantly higher than in 34.6% of DCIS and 15.6% of adjacent non-tumor tissues. Mortalin staining was closely correlated with histological grade, clinical stage, lymph node metastasis, lower disease-free survival (DFS) and overall survival (OS) rates of patients with breast cancer. Moreover, multivariate analysis suggested that mortalin emerged as a significant independent prognostic factor along with clinical stage and HER2 expression status in patients with breast cancer [147].

Therefore, both experimental and clinical data show that mortalin is involved in breast cancer pathogenesis via multiple mechanisms and seems to be a druggable target. 

### 4.4. Conclusions on above Two Sections

It follows from the above data that the major members of HSP70 subfamily, namely inducible forms of HSP70, GRP78 and GRP75 (mortalin) do promote tumorigenesis in the mammary gland and aggravate the course of breast cancer by stimulating the pathways, processes and phenomena that jointly mediate the tumor pathogenesis (see Figure 5). 

Table 2 and Table 3 summarize functions of the major members of HSP70 subfamily in breast cancer. Although it is clear that different representatives of HSP70s are often implicated in the same processes, they are strikingly indispensable since depletion of certain chaperones cannot be substituted by another closely related chaperone. In the case of HSPA5 and HSPA9, this can be explained by the fact that they are located in the ER and mitochondria and hence, cytosolic chaperones cannot substitute them. However, in the case of HSPA1 (HSP70) and HSPA2, it is difficult to explain since they are both located in the cytosol where a major homologous constitutive chaperone HSPA8 (HSC70) is also present.

Another notion emerges is that, besides their chaperone function, HSP70s can also regulate many processes which are directly unrelated to this function. For instance, they are, apparently indirectly, involved in the regulation of transcription (e.g., EMT-driving transcription factors), translation (e.g., HSPA5 in the ER), degradation/stabilization of proteins (e.g., p53), and a bunch of signaling pathways (see above Section 3 and Section 4 for details). In the case of cytosolic HSP70, its signaling functions seems to be mediated mainly by interaction with its co-chaperone BAG-3 (see [151] for review), but in the case of other HSP70s, it is not fully clear how such functions are realized in cancer cells and CSCs, although a number of cancer-related signaling axes that involve HSP70s have been described. In any event, such numerous indispensable functions of HSP70s in the tumor development and progression make them therapeutically attractive molecular targets since their suppression would affect many critical processes related to breast cancer (see next section).

## 5. Approaches to Targeting HSP70s to Fight Breast Cancer 

It is known that HSP70s are not only the promoters of tumorigenesis in mammary glands but also implicated in the mechanisms defining pathogenesis of breast cancer. Being the functional components of (epi)genetic regulation and signaling networks in malignant cells [12,20,21], HSP70s contribute to all the major manifestations of breast cancer pathogenesis such as unlimited, invasive tumor growth, EMT and formation of metastases, high resistance to therapeutics, etc. Consequently, any treatments somehow downregulating HSP70s in breast cancer cells would suppress the HSP70-dependent pathogenic mechanisms and exert some curative or tumor-sensitizing effects. The domain-based molecular organization of HSP70s and multilevel regulation of the expression and function of HSP70s (see Section 2 and Figure 1) enable multiple ways to target their cancer-promoting activities. In the present section, various approaches, models and agents are reviewed which have been used for targeting either HSP70 or GRP78, or GRP75 (mortalin) with the aim of fighting breast cancer. 

### 5.1. Targeting HSP70 in/on Breast Cancer Cells

In Section 3, we showed how the approaches with gene silencing (knockdown or knockout) techniques allowed researchers to prove the contribution of inducible HSP70 forms to the tumorigenicity of breast cancer cells. 

Apparently, similar approaches may be used to repress breast cancer cells or attenuate their resistance to various anticancer agents. For instance, the siRNA-induced downregulation of HSP70 in human breast cancer MCF-7 cells enhanced their sensitivity to pterostilbene [152]. HSPA1A knockdown by siRNA in triple negative breast cancer-derived MDA-MB-231 cells abrogated their resistance to radiation exposure [153]; this suggests a possibility of targeting the HSP70 expression to improve the action of radiotherapy against triple negative breast cancer. The *HSP70* gene silencing in breast cancer cells was also shown to impair the small extracellular vesicle-mediated intercellular delivery of HSP70 that conferred energy metabolism reprogramming and resistance to adriamycin in the tumor cell-recipients [154]. 

Specific methods of targeted delivery of siRNAs/HSP70 to breast cancer cells are currently being developed to ensure the better selectivity and efficacy. In particular, gold nanoparticles carrying siRNA against inducible HSP70 and hyaluronic acid (HA) for targeting CD44 (a receptor of HA) at the surface of triple negative breast cancer cells were applied to sensitize those cells to photo-thermal therapy modelled in vitro [155]. Using an in vivo model with murine breast cancer 4T1 cell line, Zhou et al. [156] have shown that HSP70-targeting siRNA sensitized the tumor to the TNF-related apoptosis-inducing ligand (TRAIL) therapy. In the latter study, an HA shell-protected hierarchically modular assembly formulation was constructed for the sequential releases of TRAIL into the extracellular space and siRNA/HSP70 into the cytoplasm which yielded the synergistic antitumor effects [156]. In the future, it seems quite realistic to use similar siRNA-containing nanoparticles or formulation constructs in the clinical setting for beneficial targeting HSP70s in patients’ breast tumors. 

Interestingly, the desirable inhibiting HSP70 in breast cancer cells may also be achieved by affecting the epigenetic mechanisms of regulation of the HSP70 expression/activity. Wu et al. [157] found that, in human breast cancer MCF-7 and MDA-MB-231 cells, transfection with microRNA-34a (miR-34a) downregulated histone deacetylases HDAC1 and HDAC7, thus preventing the deacetylation of HSP70 K246 and increasing the level of acetylated (inactive) HSP70; such a shift enhanced the cytotoxicity and autophagic cell death after treatments with doxorubicin, paclitaxel and cisplatin [157]. Long non-coding RNA HOX transcript antisense RNA (lncRNA HOTAIR) and microRNA-449b-5p (miR-449b-5p) were shown to regulate the translation of HSPA1A mRNA in breast cancer MCF-7 and MDA-MB-231 cells so that lncRNA HOTAIR-targeting siRNA can confer the miR-449b-5p-mediated blockade of HSPA1A accumulation in those cells thus leading to the decrease in their HSP70-dependent radioresistance [153]. As the same (positive) correlations between the expression levels of lncRNA HOTAIR, HSPA1A and high radioresistance were observed in the samples of patients’ breast tumors [153], targeting the lncRNA HOTAIR/miR-449b-5p/HSPA1A mRNA pathway seems to be a promising approach to sensitize mammary gland malignancies to radiotherapy. Thus, the cited refs [82,110,153,154,155,156,157] provide the ‘proof-of-method’ for the use of gene-silencing technologies to curatively target HSP70s in breast cancer. 

Even greater expectations are traditionally associated with ‘pharmacological’ inhibition of HSP70 in breast cancer cells, i.e., when the expression level or cancer-promoting activities of this chaperone would somehow be suppressed by a drug or natural compound. The actual task is to find or create such suitable drug. That is why many research groups currently focus their efforts on search for and trials of various agents inhibiting the expression or functions of HSP70. There are compounds which are able to somehow reduce the intracellular level of HSP70. Likewise, a number of cell-permeable HSP70-binding agents are known which can attack certain sites at the chaperone molecule and selectively inhibit its interactions with ATP/ADP, protein substrates or co-factors (see also Figure 1). Table 4 summarizes the numerous experimental data and references on small molecule inhibitors and peptide inhibitors of HSC70/HSP70 which demonstrated any hopeful effects in breast cancer-relevant models. 

The data from Table 4 provide the proof-of-principle that the inhibitory targeting HSP70 in breast cancer cells can exert the therapeutic effects. However, so far none from the inhibitors of the expression or activities of HSC70/HSP70 are either approved for clinical oncology or demonstrated encouraging results in clinical trials. Among all the agents listed in Table 4, valproic acid (an inhibitor of HDAC1 and HDAC2) seems to be the most likely candidate [167], as this drug is approved against epilepsy and some other convulsive diseases, while demonstrating antitumor activities in cancer-related models. Mawatari et al. [167] suggested that the valproic acid-induced cell cycle arrest and apoptosis in HER2-overexpressing breast cancer SKBR3 cells are due to the increase in acetylation of HSP70 that inactivates the chaperone and disrupts its cooperation with HSP90. However, valproic acid, besides the acetylated HSP70, has other cellular targets and the observed cytotoxicity may include mechanisms different from the one based on the increased acetylation of HSP70. Effects of valproic acid on breast cancer subtypes are thoroughly analyzed in a recent review [189]; the authors concluded that valproic acid is not suitable for monotherapy of breast cancer, while the curative potential of this drug in combined therapy of breast cancer seems quite likely but needs further in vivo experiments and trials. 

The main difficulty in creating an antitumor drug on the basis of inhibitors of HSP70 is, probably, a great importance of this chaperone for the viability of normal cells so that the in vivo HSP70 inhibition may result in the high toxicity and dysfunction of normal tissues and organs. Inhibitory targeting the inducible HSP70 appears to be a suitable approach in terms of anticancer therapy, as malignant cells are known to be more “addicted” to this form of the chaperone than normal cells (see Section 3.1 and [36,37,38]) and, therefore, the cytotoxic effects of such targeting may stronger be manifested in tumor cells, thus yielding the sought selectivity. However, the non-selective inhibition of HSF1-dependent expression or chaperone activity of inducible HSP70 may critically decrease the resistance of patients’ normal (tumor-free) tissues to the toxic effects of chemotherapy and concomitant pathophysiological stresses such as inflammation, ischemia, acidosis, endotoxinemia and others, so that in some cases, this may have fatal consequences. The desired progress would perhaps be achieved under the development of drugs which do not cause the total inactivation of HSC70/HSP70 but, instead, selectively inhibit its specific functions which are vitally important for cancer cells. In this respect, YM-1 [86,165] and JG-98 or its analogs [22,163,177,178,179], which inhibit HSP70–BAG3 interaction, thereby blocking growth of breast tumors, deserve serious attention. It seems likely that in the future, the creation of well-tolerated analogs of YM-1 or JG-98 with the same inhibitory and antitumor properties will provide new effective tools for treatment of patients with breast cancer. 

One more promising target to inhibit cancer-promoting functions of HSP70 seems to be its co-chaperone HSP40 (DNAJ) that stimulates HSP70 ATPase. Moses et al. [190] have demonstrated the feasibility of targeting the HSP40/HSP70 chaperone axis to treat castration-resistant prostate cancer (CRPR) that is insusceptible to antiandrogen therapy. In turn, Nitika et al. [191] focused their attention on DNAJA1 (HSP40 subfamily member A1) that was found to be upregulated in a variety of cancers. After chemogenomic screening the NIH Approved Oncology collection and testing DNAJA1 inhibitors in CRPR-related models, the researchers delineated DNAJA1 as a hub for anticancer drug resistance and suggested that DNAJA1 inhibition is a potent strategy to sensitize cancer cells to chemotherapy [191]. Although both studies [190,191] were performed on prostate cancer cells, similar approaches (i.e., targeting HSP40 to inhibit HSP40–HSP70 interaction) may be efficient for breast cancer as well. 

The other opportunity of clinical adoption of some HSP70 inhibitors against breast cancer may become the development and use of certain methods for targeted delivery of the inhibiting agent to the tumor cells in order to restrict the HSP70 inhibition-associated cytotoxic effects outside the tumor. For this purpose, the improved techniques of electroporation, antibody-based vectors and also HSP70 inhibitor-loaded microcarriers on the basis of liposomes, nanoparticles, dendrimers and others may be applied. The results obtained on breast cancer-related models with certain kinds of formulation including nanoparticles [172,173,176,188] seem hopeful, so further research work in those directions may yield a success. Moreover, the approach with targeted delivery is suitable for not only the small molecule inhibitors but also specific gene vectors constructed for the HSP70 downregulation in breast tumors. 

Given that only cancer cells expose HSP70 on their surface [25], agents targeting cell membrane-associated HSP70 may be more selective toward tumors than cell-permeable inhibitors of intracellular HSC70/HSP70. An attempt to exploit such an agent has been modeled with a 14-mer peptide (TKD) derived from the oligomerization domain of membrane HSP70 as a tumor cell-targeting ligand to modify polymeric micelles loaded with doxorubicin [192]. The TKD modification ensured the selective delivery of doxorubicin-containing micelles to human breast cancer MCF-7 cells expressing HSP70 on their plasma membrane; this TKD-mediated selective targeting was accompanied by an increased uptake of the drug-loaded micelles and retarded proliferation in the treated MCF-7 cells. The enhanced accumulation of TKD-modified micelles within the MCF-7 xenografts in nude mice proved the tumor-targeting ability of such an approach in vivo [192]. Therefore, the selective targeting HSP70 expressed on the breast cancer cell surface may be a promising alternative for the inhibition of intracellular HSC70/HSP70. 

Surprisingly, the overexpression of one of the inducible HSP70 forms, HSPA6, in breast cancer cells may somehow be associated with the repression of their malignant traits. Shen et al. [193] observed that thymoquinone (a component of black seed oil) enhances the HSPA6 expression in triple negative breast cancer-derived BT-549 cells, while inhibiting their growth, migration and invasion. Those antitumor effects of thymoquinone became attenuated in the case of knocking down of HSPA6 in BT-549 cells; in vivo, the high HSPA6 expression was found to be positively correlated with long overall survival in patients with breast cancer, thus suggesting the tumor-suppressive roles for this form of inducible HSP70 [193]. This intriguing publication needs, though, confirmation, while the implication of HSPA6 in the breast cancer pathogenesis remains to be established. In turn, thymoquinone deserves attention as a natural compound, potentially having HSPA6-dependent activities against triple negative breast cancer. 

The summation on this subsection: After well-modelled experiments in vitro and in vivo, the possibility of repressing and/or sensitizing breast cancer via inhibitory targeting HSP70 expressed in/on the tumor cells can be considered proven. That said, the state-of-the art in this direction is still far from real adopting any HSP70-inhibiting agent in anti-cancer therapy. The authors suppose that expected advances in creation of highly selective (i.e., low toxic for normal cells) inhibitors of certain HSP70—co-factor interactions along with further development/improvement of pharmacological systems for formulation and targeted delivery of therapeutic agents will help to achieve the sought-after method for curative targeting HSP70 in breast tumors. 

### 5.2. Targeting GRP78 (HSPA5) in Breast Cancer Cells

Being the master regulator of the UPR, GRP78 largely resides in the ER, while its minor part can be expressed at the cancer cell surface and also incorporated into secreted exosomes [26,28,29]. The roles of both intracellular and extracellular GRP78 in the development and pathogenesis of breast cancer have been thoroughly considered in Section 4.2. In the next two subsections, various attempts at targeting either intracellular GRP78 or cell surface-exposed GRP78 are reviewed in the context of searching for novel methods to combat breast cancer. 

#### 5.2.1. Targeting GRP78 Inside Breast Cancer Cells 

As well as in the situation with HSP70, the use of the knockdown technique allowed researchers to downregulate intracellular GRP78 in breast cancer-relevant models to achieve the tumor-repressing or tumor-sensitizing effects. Therefore, knockdown of GRP78 was shown to sensitize breast cancer MCF-7 cells to paclitaxel [122], taxol and vinblastine [194], bortezomib and panobinostat [195]. Li et al. [196] found that knockdown of GRP78 in CSCs from a breast cancer MCF-7 cell line can sensitize them to γ-radiation exposure, thus indicating a potential way to increase the efficacy of radiotherapy towards breast cancer. It was also reported that siRNA-induced knockdown of GRP78 restores the antiestrogen sensitivity in the tamoxifen-resistant breast cancer cells (MCF-7/LCC9) [137]; combining GRP78 inhibitors with the antiestrogen therapy has been suggested for better targeting breast cancer [197,198]. Later, the same research group successfully used the injections with antisense oligonucleotides targeting GRP78 (morpholino) to sensitize the MCF-7/LCC9 xenograft tumors in mice to tamoxifen [199]. Yao et al. [200] have demonstrated that siRNA-induced downregulation of GRP78 in MCF-7 cells sensitizes them to 5-fluorouracil; this effect was observed both in vitro and in the xenografts grown from MCF-7 cells pre-transfected with GRP78-targeting siRNA. In a model with human triple negative breast cancer-derived BT-549 cells, shRNA-conferred knockdown of GRP78 was demonstrated to sensitize the GRP78-depleted cancer cells to doxorubicin and cisplatin [140]. Taken together, those data [122,137,140,194,195,196,197,198,199,200] indicate a real possibility of the usage of GRP78 gene-silencing vectors against breast cancer, and as such an approach enables to sensitization of this malignancy to chemotherapy, antiestrogen therapy and radiotherapy. 

If seriously considering the therapeutic potential of the knockdown technique, it is worth thinking about methods of the delivery of siRNAs or shRNAs, or antisense oligonucleotides to human breast tumors. The experimental studies in this direction are carried out, in particular using cell-penetrating nanostructures or liposomes as carriers for GRP78 gene-silencing vectors. Patel et al. [201] described the GRP78-targeting siRNAs self-assembled into nanostructures which exerted the GRP78 knockdown-associated cytotoxic effects on triple negative breast cancer-derived MDA-MB-231 cells in vitro. A model system for co-delivery of camptothecin and GRP78-targeting siRNA was developed on the basis of 1,2-dioleoyloxy-3-trimethylammoniumpropane (DOTAP) liposomes [202]. Those DOTAP liposomes ensured the greater efficiency of transfection (as compared to the popular lipofectamine technique) and also conferred the increased sensitivity to camptothecin in breast cancer MCF-7 cells and CSCs [202]. Further progress in the creation of nanostructures and liposomes for the delivery of GRP78 gene-silencing vectors may yield new tools for targeting breast tumors. 

In parallel, approaches are currently being developed to ‘pharmacological’ targeting intracellular GRP78. Many papers report small molecule compounds which inhibit the expression or cancer-promoting functions of GRP78 in breast tumor cells, thereby exerting some beneficial effects. Table 5 summarizes such data along with the respective references. 

The data of Table 5 support the idea that pharmacological targeting GRP78 inside breast cancer cells can fight the disease. Nevertheless, among all the GRP78-inhibiting agents presented in Table 5, only panobinostat (LBH589 or farydak) is an approved anticancer drug which is orally applied against multiple myeloma. As for human breast cancer, back in 2010 Rao et al. [195] reported that panobinostat inhibits HDAC6 in cultured MCF-7 and MDA-MB-231 cells, thereby increasing the acetylation of GRP78 in 11 lysine residues. Such modification of the chaperone was shown to disrupt its complex with PERK and initiate the UPR followed by the expression of CHOP and some other pro-apoptotic proteins as well as the activation of caspase-7 in the drug-treated breast cancer cells [195]. 

If the panobinostat-induced acetylation of GRP78 does enable triggering of the apoptotic scenario in human breast cancer cells, this GRP78-mediated effect seems very attractive. However, Chang et al. [27] found that panobinostat inhibits the deacetylation and subsequent ubiquitination/degradation of HSPA5 in MDA-MB-231 cells transfected with E1A adenovirus; this inhibition resulted in the increase in cellular HSPA5 level which, according to the authors’ concept, may promote metastases and impair the antimetastatic effect of E1A expression in an in vivo murine model. In any case, additional data are required for the elucidation of the therapeutic potential of panobinostat toward breast cancer and roles of GRP78 in the drug-induced effects. Meanwhile, the phase I study of panobinostat combined with letrozole in postmenopausal metastatic breast cancer patients was rather successful and the oral doses of either drug for phase II were scheduled [216]. It is of note, however, that the panobistat effect may be unrelated to GRP78 inhibition since it is a pan-HDAC inhibitor. 

In contrast to cytosolic HSP70, whose activities are entirely aimed at cytoprotection, the GRP78-mediated induction of UPR may be either adaptive or lethal for the cell undergoing ER stress [24,26]. Besides panobinostat, other modulators of the GRP78 stability/function may be found or created which would switch the UPR pathway in breast cancer cells from cytoprotection to apoptosis. 

The summation on this subsection: In various breast cancer-relevant models, the GRP78-targeting natural and chemical compounds listed in Table 5 demonstrate rather good effects such as apoptosis in the tumor cells and sensitizing them to chemotherapeutic drugs [195,203,204,208,209,213,215], repression of the tumor growth and CSCs [127,206,207,215], inhibition of the tumor invasion and metastasis formation [127], and others. In the future, it seems doable to select among natural or newly synthesized agents a clinically applicable inhibitor of GRP78 which will be an effective remedy against breast cancer. For this purpose, further studies with GRP78 structure-based molecular docking and in silico biosimulation are, obviously, required. 

#### 5.2.2. Targeting Breast Cancer Cell Surface-Exposed GRP78 

GRP78 expressed on the cancer cell surface contributes to the maintenance of cancer stemness and plays pivotal roles in the mechanisms of EMT, tumor invasion and metastasis spread [28,29]. In particular, the breast cancer cell surface expression of GRP78 is associated with their metastatic phenotype and resistance to tamoxifen [132,217] and cisplatin [28]. On one hand, cell surface-attached GRP78 is a breast cancer-promoting factor (see Section 4.2 for details) but, on the other hand, the presence of GRP78 on the breast cancer cell surface renders this chaperone targetable for GRP78-binding agents (e.g., chemical inhibitors, peptides, proteins or antibodies) which may exert the antitumor effects. Such targeting seems especially attractive because the high level of cell surface-expressed GRP78 is a phenotypic marker of CSCs and metastasis-forming CSC-like cells in a breast tumor cell population [28,29]. Several approaches are here considered for targeting of the breast cancer cell surface GRP78 with non-antibody agents, whereas the antibody-based targeting is described in Section 6. 

The idea to use GRP78-binding peptide motifs to attack GRP78 on the breast cancer cell surface has been suggested back in 2004: Arap et al. [218] have described two synthetic GRP78-binding peptides fused to the programmed cell death-inducing domain D(KLAKLAK)2, an amphipatic α helix-forming antimicrobial peptide that disrupts eukaryotic mitochondrial membranes upon receptor-mediated internalization. Being intravenously injected in tumor-bearing mice, those chimeric peptides suppressed the growth of murine breast carcinoma EF43-fgf4, while the cancer cell surface-expressed GRP78, as the peptide-targeted receptor, ensured the cell death-driving internalization [218]. Later, one of such GRP78-binding proapoptotic peptides (designated bone metastasis-targeting peptide 78 (BMTP78)) was explored in the relevant models with murine and human breast cancer cells [128]. The tumor growth-suppressing and anti-metastatic actions of BMTP78 were revealed for murine breast carcinomas 4T1 and EF43-fgf4 in vivo as well as for human triple negative breast cancer MDA-MB-231 xenografts in mice [128]. In 2018, the toxicology studies of BMTP78 in rodents and primates were published [219], and an applicability of this peptide agent for breast cancer treatment remains to be examined in future preclinical and clinical trials. 

Using phage display biopanning, Gly-Ile-Arg-Leu-Arg-Gly (GIRLRG) has been identified as a GRP78-binding peptide that selectively recognizes malignancies responding to radiation exposure [220]. Due to cancer cell surface-expressed GRP78 being the receptor target for GIRLRG, conjugation of GIRLRG to a sustained-release nanoparticle drug delivery system yielded increased paclitaxel concentration, apoptosis and growth delay in irradiated human breast MDA-MB-231 tumor xenografts [220]. 

An original approach was tried out by Dobroff et al. [221], who found a GRP78-binding motif displayed on adeno-associated virus/phage particles and used those GRP78-targeting particles to deliver the human Herpes simplex virus thymidine kinase (HSVtk) type-1 transgene to a murine xenograft model of SUM190 human inflammatory breast carcinoma. This delivery enabled both the in vivo diagnosis through positron emission tomography (PET) imaging and selective treatment of breast carcinoma thanks to locally restricted activation of the HSVtk-sensitive prodrug ganciclovir within the tumor sites [221]. 

In another model of murine breast cancer, exogenous isthmin (a cell-secreted 60-kDa protein), was shown to bind to GRP78 on the surface of vascular endothelial cells and breast carcinoma 4T1 cells; in both types of the cells, the formed isthmin–GRP78 complex is internalized through clathrin-mediated endocytosis with subsequent induction of mitochondrial dysfunction and apoptosis [222]. In vivo, the breast tumor growth and angiogenesis were suppressed in mice intravenously injected with isthmin [222]. Kao et al. [223] reported a cyclic peptide, BC71, harboring the RKD motif in the C-terminal adhesion-associated domain of isthmin that acts as the proapoptotic ligand of cell surface-expressed GRP78. In that study, intravenous administration of BC71 suppressed the growth of breast 4T1 carcinoma in mice with inhibiting angiogenesis and enhancing apoptosis in the tumor. Being conjugated to a fluorescent label, BC71 accumulated within the 4T1 tumor in mice by binding to the tumor cell surface-exposed GRP78; such selective uptake of the labeled ligand suggests a possibility of usage of BC71 for in vivo imaging of breast cancer (e.g., with BC71-based probes for PET) [223]. However, given the fact that BC71 induces apoptosis in the vascular endothelium [223], the applicability of this peptide in theranostics should be thoroughly tested in the relevant trials. 

Another research group [224], after using homologous modeling and molecular docking, selected GRP78-binding peptides that form non-dissociating complexes with GRP78 on the cell surfaces of human carcinomas including triple negative breast cancer-derived MDA-MB-231 cells and also tumor cells isolated from clinical breast cancer specimens and xenografted in mice. In vivo, the conjugates of those GRP78-targeting peptides to liposomes with (188)Re yielded significantly greater uptake of (188)Re by the breast cancer xenografts as compared with the peptide-free (188)Re-liposomes [224]. The conjugation of those peptides to liposomes with doxorubicin also significantly enhanced the tumor growth-suppressing effect of such liposomes in the breast cancer xenograft-bearing mice. Besides the enhanced suppression of tumor growth, the doxorubicin-containing liposomes conjugated to the GRP78-binding peptides substantially diminished CSC subpopulations in the breast cancer xenografts [224]; if so, these (or similar) peptide–liposome conjugates may become a potent weapon against breast cancer stemness. Therefore, the delineated strategy with GRP78-targeting peptides may help to develop new agents for breast cancer diagnostics/imaging and treatment. 

Tseng et al. [132,217] studied a contribution of cell surface GRP78 to the membrane homeostasis, metastatic capability and viability of tamoxifen-resistant breast cancer MCF-7 cells. It was revealed that the C-terminal proline-rich region of GRP78 is critical for both GRP78 expression on the breast cancer cell surface and interactions of GRP78 with CD44v (a transmembrane protein involved in metastasis spread). The enforced (plasmid-induced) expression of a short peptide bearing the proline-rich region of GRP78 in MCF-7 cells resulted in the reduction in CD44v and cyclin D1 protein levels along with the activation of apoptotic caspase-3 [132]. Thus, the cancer-promoting functions of breast cancer cell surface-expressed GRP78 may be compromised by endogenously synthesized peptides bearing certain motifs of GRP78. In vivo, such an approach may only be realized through ‘gene therapy’ with targeted delivery of the peptide-expressing plasmid- or virus-based vectors into breast cancer cells. 

The summation on this subsection: After reviewing the above data obtained in breast cancer models, GRP78 expressed on the surface of breast tumor cells seems to be an extremely attractive molecular target for therapeutic attacking breast cancer cells and, among them, breast CSCs and metastasis-forming CSC-like cells. Besides the GRP78-affecting peptides, circulating or administered anti-GRP78 antibodies may also be used as agents targeting GRP78 on the surface of breast cancer cells (see further Section 6.2.2). However, a lot of research work should additionally be performed to develop a clinically applicable therapeutic agent targeting cell surface-expressed GRP78 in breast tumors of patients. 

### 5.3. Targeting GRP75 (HSPA9 or Mortalin) in Breast Cancer Cells 

GRP75 contributes to tumorigenesis in mammary gland and then plays cancer-promoting roles in the disease course (see Section 4.3).

Some of breast cancer-promoting mechanisms may be inhibited by targeting the GRP75 expression in breast cancer cells by means of siRNA or expressing certain genes. In particular, the GRP75 expression can be suppressed in human breast cancer cells by overexpressing the PCHGB7 gene in them: Hou et al. [225] found that PCHGB7 negatively regulates the GRP75 (HSPA9) expression in triple negative breast cancer HS578T and BT549 cells. As the enforced PCHGB7 expression in those cells resulted in the induction of caspase-dependent apoptosis and sensitization to carboplatin [225], this finding delineates PCHGB7 as a potential tool for targeting GRP75 in breast tumors. 

Theoretically, the GRP75-targeting ‘gene therapy’ may exert some curative or tumor-sensitizing effects in patients with breast cancer. Of course, to adopt such an approach in the clinical setting, special methods of the in vivo delivery of those (cell-transfecting) vectors to the target tumors should be developed and tested in the relevant trials.

As a realistic alternative to the GRP75-targeting ‘gene therapy’, the cancer-promoting functions of GRP75 can be inhibited by treating breast tumor cells with cell-permeable drugs or peptides which somehow downregulate the GRP75 expression/activity. These approaches have been published and Table 6 presents the data/refs from several attempts to target GRP75 in order to fight breast cancer. 

As can be seen from the data of Table 6, it is the GRP75 (mortalin)–p53 complex formation that is affected by various GRP75-targeting agents which have the antitumor activity. The achieved abrogation of mortalin–p53 complex appears to liberate p53 and thus activate its transcriptional function promoting cell cycle arrest and apoptosis. This mechanism is, probably, not a universal one: despite abrogating the mortalin–p53 interaction in both MCF-7 and T47D cells, mortaparib^Plus^ did not induce apoptosis in T47D cells [232]. The researchers explained this discrepancy by the inability of mortaparib^Plus^ to disrupt the apoptosis-inducing factor (AIF)–mortalin complexes and suggested the AIF-mediated (i.e., caspase-independent) mechanism of cell death in breast cancer T47D cells [232]. If so, these AIF–mortalin complexes may also be the therapeutically attractive target under the development of novel antitumor drugs on the basis of molecular docking. 

The summation on this subsection: Having examined the presented data and looking ahead, one can expect that further exploration of mortalin–p53 interactions supported by studies with in silico modeling and biosimulation will eventually yield suitable HSPA9-targeting agents which will be useful for treating patients with breast cancer. On the other hand, as a number of natural (largely plant-derived) compounds produced the sought-after activities [227,229,230], it cannot be ruled out that the suitable agent someday will be discovered among natural products to curatively target mortalin in human mammary gland malignancies. 

## 6. HSP70s as Potential Targets or Tools for Immunotherapy of Breast Cancer

In the last decade, anticancer immunotherapy has shown great advances and the same trend is predicted in the future [233]. While the best results of immunotherapy took place under treatment of melanomas, making this modality equally effective against breast cancer is an urgent task for modern oncology. The present section reviews various approaches to use of HSP70s (and their peptide derivatives) as either targets for antibodies or adjuvants and vaccines for inducing/enhancing the antitumor immune response to fight breast cancer. 

### 6.1. Inducible HSP70 as a Potential Target or Tool for Immunotherapy of Breast Cancer 

#### 6.1.1. Released and Cancer Cell Surface-Exposed HSP70 

Bausero et al. [234] found that interferon-γ stimulates active release of exosomes enriched by inducible HSP70 in murine breast cancer 4T1 cells; in turn, those HSP70-containing exosomes upregulated the CD83 expression and stimulated interleukin-12 release by naive dendritic cells that suggested the HSP70-dependent enhancement of immune surveillance over the tumor. Besides interferon-γ [234], the antitumor response-promoting release of HSP70-enriched exosomes by breast cancer cells may also be stimulated by mild hyperthermia [235]. It was shown that exposure of naive murine macrophages to such exosomes released from preheated breast cancer 4T1 and EMT-6 cells resulted in an increased expression of specific macrophage activation markers [235]; this finding suggests the immunogenic potential intrinsic in the HSP70-containing exosomes released from breast cancer cells treated with mild hyperthermia. 

The inducible HSP70 expressed on the surface of breast cancer cells can be targeted by specific anti-HSP70 antibodies conjugated to antitumor agents. For example, Cheng et al. [236] fabricated a microwave-triggered, HSP70-targeted gold nano-system with a gold nanocage as a photothermal conversion agent and a monoclonal antibody to HSP70 as a targeting ligand. Those HSP70-targeting nanoparticles demonstrated good antitumor effects in a murine model of photothermal therapy with 4T1 breast carcinoma [236]. 

Further investigations will help to further establish whether extracellular (exosomal) and cell membrane-associated HSP70 can be exploited for treatment of patients with breast cancer. 

#### 6.1.2. HSP70-Peptide Complexes and HSP70-Derived Peptides as Adjuvants or Enhancers of Immune Response against Breast Cancer 

There were ideas and attempts to use peptide fragments from HSP70 for the preparation of vaccines against breast cancer. Back in 2004, Faure et al. [237] described the p391 and p393 peptide sequences from the molecule of human inducible HSP70 that have a high affinity for the major histocompatibility complex HLA-A*0201. Being able to trigger the cytolytic T-lymphocyte (CTL) activation, those HSP70 epitopes were shown to be the targets of an immune response in many HLA-A*0201+ breast cancer patients. The researchers suggested that HSP70, as a tumor antigen, and the HSP70-derived peptides p391 and p393 can be used to enhance the immune response/CTL attack against breast and other tumors [237]. A 2019 paper [238] reported the identification of a multi-HLA-class I–binding (promiscuous) HSP70-derived epitope peptide that binds to HLA-A*0201, *0206, and *2402 and may be applicable to cancer immunotherapy in patients with HLA-A*2402+, *0201+, and *0206+ HSP70-expressing tumors. In that study [238], an ability of the promiscuous HSP70-derived epitope peptide to induce the antitumor response has been demonstrated only for hepatocellular carcinoma but this peptide may also act against breast tumors expressing the respective HLA-A subclasses. 

Another approach has been employed by Kim et al. [239]: human HSP70 was fused to the extracellular domain of rat HER2/Neu (NeuEDHSP70) to enhance anti-tumor immunity in murine breast tumor models with 4T1.2-Neu cells and TUBO cells. The NeuEDHSP70 DNA vaccine induced the Neu-specific antibody generation and cellular immune responses in vivo that significantly increased survival and reduced metastasis in mice with aggressive spontaneous metastatic breast tumors [239]. Later, similar approaches based on HSP70 associated with tumor cell-derived proteins or peptides were used in the relevant models to enhance the immune response against breast cancer. 

Gao et al. [240] developed a method of isolation of HSP70–peptide complexes from HER2-overexpressing human breast cancer SKBR-3 cells. The obtained protein/peptide fraction was enriched by the HSP70–HER2 complexes. Dendritic cells (DCs), after being pulsed with the isolated HSP70–HER2–peptide complexes, induced the most specific CD8+ T cells that selectively killed the breast cancer cells [240]. Other researchers proposed to use HSP70–peptide complexes from DCs fused to tumor cells [241,242]. Such complexes, being extracted from DCs fused to established human breast cancer cells, demonstrated the improved effects as a component of the antitumor vaccine. In particular, the HSP70–peptide complexes from the fused DC–tumor hybrid cells exhibited the stronger immunogenicity and augmented killing of tumor cells as compared with the effects of HSP70–peptide complexes from the unfused (tumor) cells [241]. Developing the same idea, Zhang et al. [243] have isolated HSP70–peptide complexes from DCs fused to murine breast cancer 4T1 cells and then encapsulated those complexes into nanoliposomes to improve their stability/bioavailability and enhance the antitumor immunity. The antitumor vaccine based on such liposomes did enhance the antitumor immunity that was manifested in the increased T-cell activation/CTL response and elevated breast tumor therapy efficiency as compared with the control (liposome-free) variant [243]. 

A different approach with HSP70–peptide complexes was examined by another research group [72]: the A8 peptide aptamer that binds to the extracellular domain of membrane HSP70 and targets HSP70-expressing exosomes from patients with malignancies including breast cancer has been used. The researchers found that HSP70 exposed on the membrane of cancer cell-generated exosomes interacts with Toll-like receptor 2 (TLR2) on the surface of myeloid-derived suppressive cells (MDSC), thereby activating the latter and protecting cancer cells from immune recognition/attack. The HSP70-binding A8 peptide aptamers were shown to form complexes with the exosome-expressed HSP70 and block its interaction with TLR2, thus preventing the exosome-mediated MDSC activation as well as the MDSC-conferred immune evasion of cancer cells [72]. The ability of A8 to restore the in vivo antitumor immune response was demonstrated only in a murine model with melanoma B16 [72]; however, the same approach may similarly be effective towards breast cancer. 

Despite the encouraging results obtained in the above experimental models, further thorough studies are required to create the effective HSP70-based vaccines and adjuvants for immunotherapy of breast cancer. 

### 6.2. GRP78 (HSPA5) as a Potential Target for Immunotherapy of Breast Cancer 

Both intracellular (ER-localized) GRP78 and cell surface-expressed GRP78 play disease-promoting roles in the development/course of breast cancer (see Section 4.2) and are thought to be potential targets for treating or sensitizing mammary gland malignancies with certain GRP78-affecting agents (see Section 5.2). The present subsection reviews various approaches to targeting either intracellular or cell surface GRP78 in order to treat breast cancer by immunotherapy. 

#### 6.2.1. Targeting GRP78 Inside Breast Cancer Cells to Enhance the Antitumor Immune Response 

The technique of knockdown was used to prove the possibility of inducing the immune response against breast tumor cells by targeting the GRP78 expression in them. Cook et al. [199] administered the GRP78-targeting morpholino to abrogate the resistance to tamoxifen in the MCF-7/LCC1 tumor xenografts; in the morpholino-injected mice, the researchers found the elevated serum levels of monocyte chemotactic protein 1 along with the reduced CD47 expression and augmented macrophage infiltration in the GRP78-depleted tumor areas. The role of tumoral GRP78 in tumor evasion of the innate immunity has been suggested [199]. It was later shown that the GRP78 downregulation in RNAi-treated murine breast cancer 4T1B cells enhances their elimination via macrophage clearance [140] (see Section 4.2.3). Another publication of the same research group reported that RNAi-conferred knockdown of GRP78 decreases the CD47 expression in tamoxifen-resistant MCF-7/LCC9 cells and prevents the tamoxifen-mediated CD47 induction in tamoxifen-sensitive MCF-7/LCC1 cells [244]. Since CD47 is the “do not eat me” signal for macrophages, this effect could promote the innate anti-tumor immunity. 

Another breast cancer cell surface-expressed protein, namely PD-L1, is accountable for the tumor evasion of immune attacks (see Section 4.2.3) and regulated by GRP78 [140]. Importantly, knockdown of GRP78 in shRNA-treated BT-549 cells resulted in the significant decrease in PD-L1 on their surface [140]; this means that enforced inhibiting the GRP78 expression in triple negative breast cancer cells can render them more vulnerable to attacks of cytolytic T-cells. 

Taken together, these data [140,199,244] support the idea of targeting GRP78 inside breast cancer cells to stimulate the immune response against the mammary gland malignancies. This is the case when ‘gene therapy’ with GRP78-targeting siRNA- or shRNA-, or morpholino-based vectors may assist immunotherapy. However, the clinically applicable methods of such GRP78-targeting ‘gene therapy’ remain to be developed and tested in patients. 

It is strange that there are no publications about the enhancement of immune response against breast cancer by small molecule agents which are able to suppress the expression or activities of GRP78 in the tumor cells. Such a GRP78-affecting agent as nifetepimine (a dihydropyrimidone) was shown to manifest the antitumor properties in models of triple negative breast cancer (see Table 5 and refs [207,245]) and promote antitumor immune response by protecting CD4(+) T cells from tumor-induced apoptosis [246], but the potentially beneficial effect of nifetepimine on the immune response was not related to targeting tumoral GRP78. Meanwhile, it seems likely that certain inhibitors of the GRP78 expression/activities in breast cancer cells would help to pharmacologically enhance the antitumor immune response and could be combined with immune checkpoint inhibitors to better treat mammary gland malignancies. The latter suggestion, though, needs special verification. 

#### 6.2.2. Targeting Breast Cancer Cell Surface-Exposed GRP78 with Anti-GRP78 Antibodies 

GRP78 expressed on the tumor cell surface contributes to the pathogenesis of breast cancer (see Section 4.2.2) and appears to be a very appropriate target for selective attacking with GRP78-recognizing immunoglobulins. Indeed, no benign cells but only malignant ones and especially CSCs expose GRP78 on their surface [28,29], so that specific anti-GRP78 antibodies (and conjugates of such antibodies to antitumor agents) can selectively target breast cancer cells without affecting normal tissues. 

Semiconductor quantum dots (Qdots), being conjugated to small antibody fragments targeting cell membrane-bound GRP78, were shown to bind to the surface of human breast cancer MDA-MB-231 cells and then become internalized; importantly, those Qdot conjugates inhibited the MDA-MB-231 xenograft growth in nude mice [247]. This study [247] presented an example of how conjugates of anti-GRP78 antibodies with an antitumor agent can successfully target triple negative breast cancer. 

A novel methodology, termed selection of phage-displayed accessible recombinant targeted antibodies (SPARTA) has been applied to generate human antibodies targeting GRP78 on the breast cancer cell surface [248]. In combination with secondary antibodies conjugated to such toxins as α-amanitin or duocarmycin, the anti-GRP78 antibody promoted killing of murine breast cancer EF43fgf4 cells [248]. 

Interestingly, unconjugated antibodies targeting GRP78 on the breast cancer surface can exert beneficial effects as well. For instance, it was reported that binding of anti-GRP78 antibodies to GRP78 on the surface of breast cancer MCF-7 cells suppresses their proliferation and migration mediated by the STAT3 phosphorylation [131]. Tseng et al. [217] have revealed that antibodies targeting GRP78 on the surface of tamoxifen-resistant breast cancer MCF-7-LR cells can effectively reduce the cell surface expression of CD44v and cell spreading, thus suggesting a possibility of the attenuation of metastatic potential of the tumor by anti-GRP78 antibodies. Later, it was shown on GRP78-overexpressing MDA-MB-231 cells that in vitro treatment with a specific antibody targeting GRP78 on their surface decreased cell viability and sensitized these cells to cisplatin [28]. 

Therefore, the studies described in refs [28,131,247,248] delineate antibodies recognizing GRP78 on the cancer cell surface as a promising tool for selective targeting of breast malignancies. 

### 6.3. GRP75 (Mortalin) as a Potential Target for Immunotherapy of Breast Cancer Cells 

This member of the HSP70 subfamily is known to contribute to the development and pathogenesis of breast cancer (see Section 4.3). In the present subsection, GRP75 (HSPA9 or mortalin) is considered as a molecular target for inducing the complement-dependent cytotoxicity toward breast cancer cells [231,249] and also for antibody-mediated delivery of the mortalin inhibitor CAPE into breast cancer cells [250]. 

Huang et al. [231] used a series of peptides derived from the Secretion Modification Region (SMR) of HIV-1 Nef protein to antagonize mortalin in breast cancer and leukemia cells (see also Table 5). They showed that some of the SMR-derived peptides could inhibit the expression of both mortalin and complement C9 in human breast cancer MDA-MB-231 and MCF-7 cells. Importantly, those SMR peptides abrogated the mortalin-mediated ability of breast cancer cells to release extracellular vesicles (exosomes), thereby blocking the mortalin-associated protection of the cancer cells from complement-dependent cell death [231]. Such treatments with mortalin-affecting SMR peptides are able to re-establish the complement-mediated cytotoxicity toward breast cancer cells and sensitize the latter to antibody/complement-based immunotherapy. 

Another research group has created mortalin-mimetic peptides with amino acid sequences predicted to be involved in the interactions of mortalin with its client proteins [249]. Two of those peptides, namely Mot-P2 and Mot-P7, were found to induce the cytotoxicity in malignant cells of different origin including human breast cancer T47D cells. In a respective model, both Mot-P2 or Mot-P7 were shown to significantly enhance antibody-mediated and complement-dependent cell killing in the peptide-treated cancer cells [249]. Therefore, the reported mortalin-mimetic peptides may be combined, as adjuvants, with complement-activating antibody therapy to better target breast cancer. Both relevant studies [231,249] characterize mortalin (GRP75) as a peptide-sensitive molecular target enabling eradication of breast cancer cells via complement-dependent cytotoxicity. 

Wang et al. [250] exploited a unique cell internalizing anti-mortalin antibody to generate mortalin-targeting nanoparticles loaded with CAPE (a natural inhibitor of mortalin–p53 interaction [230]). Thanks to that antibody, such CAPE-containing nanoparticles became targeted to mortalin exposed on the surface of cells of human breast carcinomas and other malignancies, then internalized and exerted cytotoxic effects [250]. Thus, specific antibodies recognizing (and binding to) mortalin at the cancer cell surface may represent one more immunotherapeutic tool for selective targeting mammary gland tumors. 

The summation on Section 6: Taken together, the cited results of model studies give the proof of concept of the applicability of HSP70s as either targets or tools for immunotherapy of breast cancer. It seems likely that this HSP70-based immunotherapy is able to enhance the antitumor action of immune checkpoint inhibitors, if these exposures are combined. However, a lot of additional experiments and, especially, clinical trials remain to be performed in order to someday introduce HSP70-based immunotherapy for treatment of patients. 

## 7. General Conclusions and Perspectives

From this review, it is easy to conclude that, on one hand, HSP70s do promote the development and pathogenesis of breast cancer (see Figure 2, Figure 3, Figure 4 and Figure 5), but, on the other hand, HSP70s may be molecular targets or beneficial tools (adjuvants) for treating human mammary gland malignancies (see Figure 6). 

Indeed, with the exception of HSPA6 [193] (whose role remains to be elucidated), the major members of HSP70 subfamily are, first, the endogenous promoters of tumorigenesis in the mammary gland and, second, tightly involved in the mechanisms and processes deteriorating the course/outcome of breast cancer. The critically important question is whether it is possible to target HSP70s in order to prevent an occurrence of breast cancer or successfully treat this disease. The studies cited here suggest that yes, theoretically it is possible. It seems quite likely that the revealed causal mechanisms of interplay of certain HSP70s with oncogene products responsible for the malignant transformation of the mammary gland epithelium as well as with regulators and components of signaling pathways that ensure the invasive, metastatic and therapy-resistant growth of breast tumors will allow researchers to develop new ways and agents for disrupting this interplay in order to prevent (or at least minimize a risk of) the HSP70-mediated tumorigenesis or alleviate the disease course/outcome. 

It appears from the present review that breast malignancies may be targeted with small molecule (natural or synthetic) inhibitors of HSP70s, antisense or si/shRNA-vectors knocking down the HSP70 expression, HSP70-binding peptides, HSP70-downregulating nanoparticles, anti-HSP70 antibodies and their conjugates, or HSP70-based vaccines (see Figure 6). In addition, certain HSP70-binding molecules and peptide derivatives of HSP70s may be used for in vivo targeted delivery of antitumor agents or tumor-marking labels to breast cancer cells in order to increase the efficacy/selectivity of the cytotoxic action towards the tumors or for tumor imaging. All those approaches demonstrated good results in various experimental models of breast cancer treatment which give rise to optimism about the possible adoption of some of these approaches (or their advanced modifications) in the clinical setting. Of course, it is unlikely that HSP70-targeting therapy or HSP70-based therapy will become a universal remedy, a kind of ‘silver bullet’, against breast cancer in humans. Rather, such HSP70-involving treatments would be helpful as adjuvant therapy in combination with conventional methods of antitumor therapy to improve patient outcomes. 

## Figures and Tables

**Figure 1 cells-10-03446-f001:**
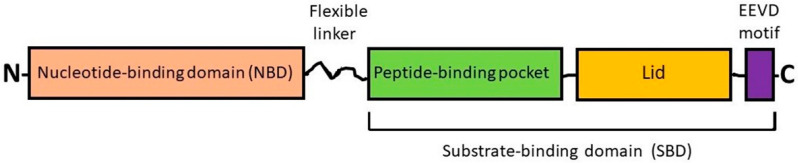
Simplified scheme showing the domain organization of HSP70s. The members of the HSP70 (HSPA) subfamily consist of two highly conserved functional domains such as the N-terminal nucleotide-binding domain (NBD) and the C-terminal substrate-binding domain (SBD) with a flexible linker between, and also have an EEVD motif at the C-terminus. The NBD contains the ATP/ADP pocket that binds ATP for the ATPase reaction. The SBD contains two subdomains: (i) a peptide-binding pocket that interacts with polypeptides as substrates and (ii) an α-helical subdomain from the C-terminal side that forms the so-called “lid”. The EEVD motif is involved in the interprotein interactions with co-chaperones and other HSPs [12,13].

**Figure 2 cells-10-03446-f002:**
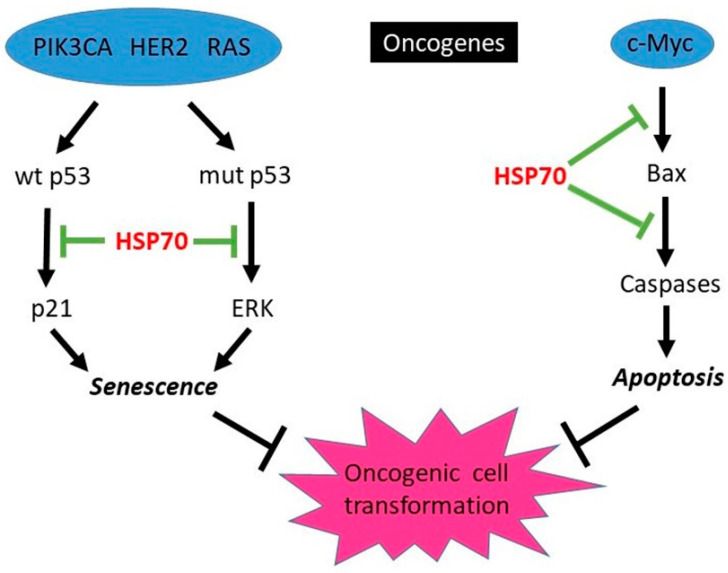
Scheme showing presumed roles of HSP70 in promotion of oncogenic cell transformation. HSP70 overexpression can suppress oncogene-induced p53-dependent and independent senescence; it can also block myc-induced apoptosis thus leading to the cell transformation.

**Figure 3 cells-10-03446-f003:**
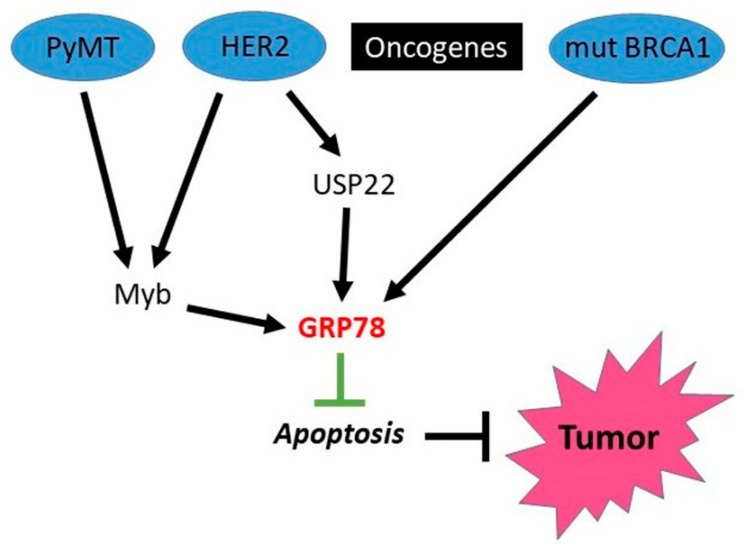
Possible mechanisms of tumorigenesis mediated by intracellular GRP78. (See text for detailed explanation.).

**Figure 4 cells-10-03446-f004:**
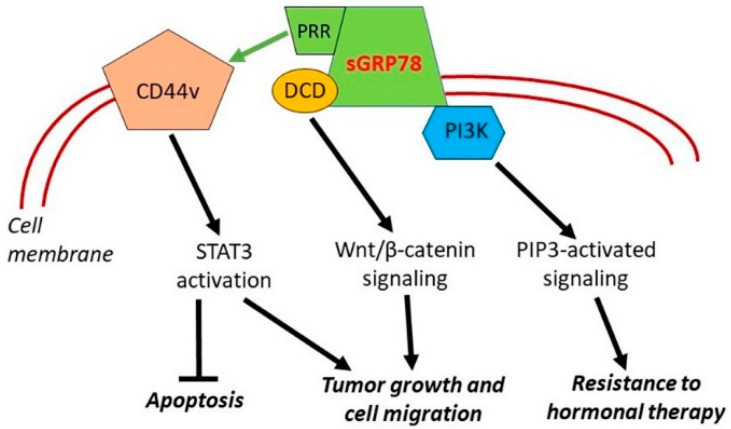
Involvement of cell surface GRP78 (sGRP78) in formation of malignant phenotype under breast cancer development. Abbreviations used: DCD—dermcidin; PRP—proline-rich region. (See text for detailed explanation).

**Figure 5 cells-10-03446-f005:**
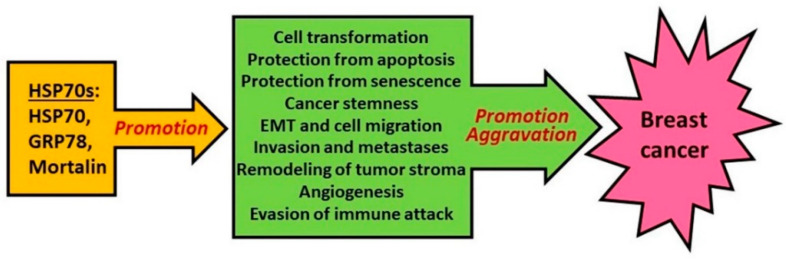
Simplified scheme showing the multilevel involvement of major HSP70s in the development and pathogenesis of breast cancer.

**Figure 6 cells-10-03446-f006:**
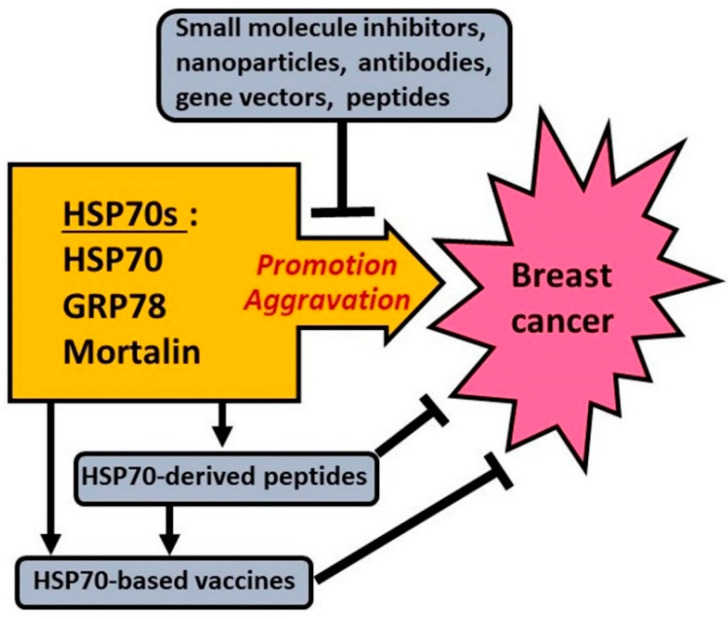
A simplified scheme showing various HSP70-exploiting approaches to targeting breast cancer.

**Table 1 cells-10-03446-t001:** Relationship between HSPA mRNA expression in subtypes of breast cancer and its estrogen receptor (ER), progesterone receptor (PR), and human epidermal growth receptor 2 (HER2) status.

HSP	Subtypes	ER+/PR+	ER−/PR−	HER2+	HER2−	Refs
HSP1A	−	+	−	−	−	[10]
HSPA2	Luminal A, B	+	−	−	+	[10,11]
HSPA5	Basal	−	−	−	−	[11]
HSPA6	Basal	−	−	−	−	[11]
HSPA8	−	−	−	+	−	[10]

+ positive correlation; − no correlation.

**Table 2 cells-10-03446-t002:** Tumorigenic properties of members of HSP70 subfamily in breast cancer.

	SUPPRESSION		PROMOTION		PROMOTION	
	Apoptosis	Senescence	Angiogenesis	EMT	Migration/Invasion	Metastases
HSPA1	[73,74]	[59,60]	[86,87,88,89,90,91]	[93]	[33,83,94,95,96,97]	[63,102,103]
HSPA2	[110]	[108]	ND	ND	[110,111,112] *	ND
HSPA5	[117]	ND	[117,129]	[126,127]	[27,123,124,125]	[27,28,123,124,127,128]
HSPA9	ND	ND	ND	[33,34,145]	[33,34,145]	ND

ND—not determined; * —no correlation was observed in ref [111].

**Table 3 cells-10-03446-t003:** Clinical significance of members of the HSP70 subfamily in breast cancer.

	Overexpression	Progression/Grade	Metastases	Mortality
HSPA1	[101,148,149]	[71,148,149]	[100,101,102]	[100,150]
HSPA2	[107,108]	NC [110]	ND	[112]
HSPA5	[113,114,115]	[101,123]	[123,131]	[141,142]
HSPA9	[147]	[147]	[147]	[147]

NC—no correlation.

**Table 4 cells-10-03446-t004:** Agents targeting intracellular HSC70/HSP70 and their effects on breast cancer.

Agent	Cells or Tumor	Molecular Target	Achieved Effects
BAG1-derived peptides [158]	MCF-7 and ZR-75-1 cells	HSC70–BAG-1 interactions	Inhibition of cell proliferation
Epigallocatechin-3-gallate [159]	MCF-7 cells,	HSP70	Inhibited HSP70 expression,
	xenografts	expression	reduced tumor size
VER155008 [103,160,161,162,163]	BT474, MCF-7 and MDA-MB-231 cells	HSP70 ATPase	Induction of apoptosis, damage of mitochondria, sensitization to TNF, heating and gemcitabine
Sulphoraphane [164]	MDA-MB-231	HSP70 expression	Downregulation of HSP70 and HSP90, apoptosis
	and MCF-7 cells		
YM-1 [86,165]	MCF-7 xenografts, MDA-MB-231 and BT-549 cells	HSP70–BAG-3 interactions	Inhibited xenograft growth,
			sensitization to drug-induced apoptosis
HS-72 [166]	BT474 and MCF-7 cells,	HSP70–ATP	Antiproliferative activity, reduced tumor size
	MMTV ^1^-neu model	affinity	
Valproic acid [167]	SKBR3 cells	HDAC ^2^,	Increased HSP70 acetylation, cell cycle arrest, apoptosis
		acetylated HSP70	
Piperidine derivatives [168]	BT474, BT/Lap(R)1.0, MDA-MB-231 and other cell lines	HSP70 ATPase	Inhibited cell proliferation, sensitization to lapatinib
Monobenzyltin complex C1 [169,170]	MCF-7, MDA-MB-231 cells and breast CSCs	HSP70 expression	Decreased HSP70 level, induction of apoptosis
Crocin [171]	MDA-MB-468 cells	HSP70 expression	Decreased HSP70 and HSP90 levels, induction of apoptosis
Gold NPs ^3^ [172,173]	MCF-7 cells	HSP70 expression	Downregulation of HSP70 and ribosome biogenesis, thermosensitization
Disubstituted thiourea [174]	BT474 cells	HSP70 ATPase	Sensitization to lapatinib
Azacytidine [175]	MCF-7 cells	HSP70 expression	Sensitization to doxorubicin
Peptide aptamers with high affinity to HSP70 (as components of NPs with doxorubicin) [176]	MDA-MB-468 cells, xenografts	Tumoral HSP70	Tumor regression, sensitization to doxorubicin
JG-98 [22,163,177,178]	MDA-MB-231 and MCF-7 cells	HSP70 allosteric site in NBD ^4^, HSP70–BAG-3 interactions	Loss of c-IAP1 ^5^ and XIAP ^6^, apoptosis and necroptosis, sensitization to drugs
JG-231 [179]	MCF-7, MDA-MB-231	HSP70 allosteric site in NBD	Cell death, reduced tumor burden in xenografts
	cells and xenografts		
MKT-077 and its analog JG-237 [180]	MDA-MB-231	“Loop 222” in	Antiproliferative
	and MCF-7 cells	NBD of HSP70	activities
PES ^7^ (or pifithrin-μ) [22,181]	MDA-MB-231and MCF-7 cells, xenografts	HSP70 SBD ^8^	Loss of c-IAP1 and XIAP, apoptosis, sensitization to photothermal therapy
Neutral analogs of JG-98 [182]	MCF-7 cells	HSP70 allosteric site	Antiproliferative activities
Artesunate [183]	4T1 and MCF-7 cells	HSP70 ATPase, HSP70 expression	Inhibition of HSP70, induction of apoptosis
Aptamer peptide conjugates [184]	MCF-7 cells	Tumoral HSP70	Sensitization to doxorubicin
Benzo-fused rhodacyanines [185]	Breast cancer cells	HSP70 chaperone Function	Antiproliferative activities, downregulation of client antiapoptotic proteins
Evodiamine [186]	MDA-MB-231 cells, CSCs, PDX ^9^ model	HSP70 NBD, HSP70 expression	Degradation of HSP70, inhibited cell proliferation, reduced tumor growth
MAL3-101 [187]	Cell lines from TNBC ^10^ and luminal subtypes	HSP70 ATPase	Induction of UPR and cell death via apoptosis
Apoptozole (as one of components of nano-diamond-based nano-platform) [188]	MDA-MB-231 cells and xenografts	HSP70 ATPase, HSP70 expression	Sensitization to photothermal chemo-combined therapy, inhibition of autophagy

^1^ Mouse mammary tumor virus (MMTV); ^2^ Histone deacetylase (HDAC); ^3^ Nanoparticles (NPs); ^4^ Nucleotide-binding domain (NBD), see Figure 1; ^5^ c-inhibitor apoptosis protein 1 (c-IAP1); ^6^ X-linked inhibitor of apoptosis protein (XIAP); ^7^ 2-phenylethynesulfonamide (PES) or pifithrin-μ; ^8^ Substrate-binding domain (SBD), see Figure 1; ^9^ Patient-derived xenografts (PDX) growing in nude mice; ^10^ Triple negative breast cancer (TNBC).

**Table 5 cells-10-03446-t005:** Agents targeting intracellular GRP78 and their effects on breast cancer.

Agent	Cells or Tumor	Molecular Target	Achieved Effects
Epigallocatechin gallate [203,204]	MDA-MB-231 and T47D cells	GRP78 ATPase	Sensitization to etoposide and quercetin, apoptosis
Panobinostat (LBH589) [195]	MDA-MB-231	HDAC6 ^1^, acetylated GRP78	Induction of UPR and apoptosis
	and MCF-7 cells		
Fukoidan [205]	MDA-MB-231 cells	GRP78 expression	Downregulation of GRP78 and apoptosis
Isoliquiritigenin [206]	MCF-7 and MDA-MB-231	GRP78 ATPase, GRP78/β-catenin/ABCG2 ^2^ signaling	Sensitization to epirubicin in vitro and in vivo
	cells, sorted CSCs, xenografts from CSCs		
Nifetepimine [207]	MDA-MB-468 and MDA-MB-231 cells, xenografts	GRP78 expression	Attenuated GRP78 induction, apoptosis, reduced tumor growth
VER155008 [208]	MDA-MB-231 and MCF-7 cells	GRP78 ATPase	Sensitization to tamoxifen, apoptosis
Plumbagin [209]	MCF-7 and T47D cells	GRP78 expression	Downregulation of GRP78, apoptosis, sensitization to tamoxifen
INH7 [210]	MCF-7 cells	17β-HSD7 ^3^, GRP78 expression	Downregulation of GRP78, apoptosis
VH1019, VH1011 [211]	MCF-7 cells	GRP78 structure- based docking	Antiproliferative and cytotoxic effects
Neoisoliquiritigenin [212]	Breast cancer cells and xenografts	GRP78 ATPase, GRP78/β-catenin signaling	Inhibition of proliferation, apoptosis
Betulinic acid [213]	MDA-MB-231 and MCF-7 cells	GRP78 ATPase	Sensitization to taxol, apoptosis
HHQ-4 [214]	Glucose-deprived breast cancer cells	GRP78 expression	Downregulation of GRP78, inhibition of proliferation
Indolylkojil methane analog (IKM5) [127]	MDA-MB-231, MDA-MB-468, MCF-7 and 4T1 cells (in mice)	GRP78 SBD ^4^, GRP78–TIMP-1 ^5^ interactions	Inhibited expression of EMT markers, suppression of invasion, tumor growth and lung metastases
Ai Du Qing formula [215]	MDA-MB-231, MCF-7 cells, breast CSCs and xenografts	GRP78 expression, GRP78/β-catenin/ /ABCG2 axis	Downregulation of GRP78, β-catenin degradation, repressing and chemo-sensitizing effects on cancer cells, CSCs and xenografts
HA15 [121]	HCC1954 and SKBR3 cells	GRP78 ATPase	Apoptosis, suppressed cell proliferation

^1^ Histone deacetylase 6 (HDAC6); ^2^ ATP-binding cassette transporter G2 (ABCG2); ^3^ 17β-hydroxysteroid dehydrogenase type 7 (17β-HSD7); ^4^ Substrate-binding domain (SBD), see Figure 1; ^5^ Tissue inhibitor of metalloproteinase-1 (TIMP-1).

**Table 6 cells-10-03446-t006:** Agents targeting GRP75 (mortalin) and their effects on breast cancer.

Agent	Cells or Tumor	Molecular Target	Achieved Effects
p53 carboxyl-terminus peptides [226]	MCF-7 cells	Mortalin–p53 interaction	Disruption of the p53–mortalin
			complex, activation of p53, growth arrest
Withanone [227]	MCF-7 cells	Mortalin–p53 interaction	Abrogation of the p53–mortalin
			complex, activation of p53, growth arrest or apoptosis
MKT-077 [227,228]	MCF-7 cells	p53-binding region of mortalin	Mitochondrial toxicity, selective killing of tumor cells
Embelin [229]	MDA-MB-231 and MCF-7 cells	Mortalin–p53 interaction, mortalin expression	Abrogation of the p53–mortalin
			complex, activation of p53, growth arrest and inactivation of metastatic signaling
CAPE ^1^ and its complex with γ-cyclodextrin [230]	MDA-MB-231 and MCF-7 cells	Mortalin–p53 interaction	Disruption of the p53–mortalincomplex, activation of p53, growth arrest, suppression of metastases
PEG-SMRwt-CLU ^2^ [231]	MDA-MB-231 and MCF-7 cells	Exosome secretion, mortalin (suggested)	Blockade of exosome release, growth arrest
Mortaparib ^Plus 3^ [232]	MCF-7 cells	Mortalin–p53 interaction	Abrogation of the p53–mortalincomplex, activation of p53, growth arrest, apoptosis

^1^ Caffeic acid phenethyl ester (CAPE); ^2^ A peptide derived from the Secretion Modification Region (SMR) of HIV-1 Nef protein that was conjugated to polyethylene glycol (PEG) on the N-terminus and to a clusterin (Clu)-binding peptide on the C-terminus; ^3^ (4-[(1E)-2-(2-phenylindol-3-yl)-1-azavinyl]-1,2,4-triazole), a novel synthetic small molecule named Mortaparib ^Plus^.

## Data Availability

Data sharing is not applicable to this article.

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
