# Peer review of "HSP70s in Breast Cancer: Promoters of Tumorigenesis and Potential Targets/Tools for Therapy"

_cells, 2021, doi:10.3390/cells10123446_

Round 1

Reviewer 1 Report

This review by Kabakov et al., discusses the HSP70-mediated cellular mechanisms and pathways  contributing to the oncogenic transformation of mammary gland epithelium and growth of  breast carcinomas. These carcinomas are invasive, metastatic and are highly resistant to host immunity  and conventional therapeutics. HSP70 isoforms are  potential targets for therapy of breast cancer.  Approaches such as HSP70-inhibitors HSP70-binding antibodies and HSP70-based vaccines have been discussed in great details.

Here are my minor concerns:

In the introduction section, please discuss the numbers and statistics of breast cancer patients worldwide.

In section 2, please add some discussion about oligomerization of Hsp70 and cite : Takakuwa et al.,Oligomerization of Hsp70: Current perspectives on regulation and function.

Figure 2 is crude and can be improved using cartoon depiction of the pathways that authors want to draw attention to.

In line 915 "The main difficulty in creating an antitumor drug on the basis of inhibitors of HSP70 .... organs.", authors should discuss alternative strategies such as targeting co-chaperones in other cancer types, as discussed in Moses et al. Targeting the Hsp40/Hsp70 Chaperone Axis as a Novel Strategy to Treat Castration-Resistant Prostate Cancer and Nitika et al., Chemogenomic screening identifies the Hsp70 co-chaperone DNAJA1 as a hub for anticancer drug resistance.

Author Response

Authors’ responses to Reviewer 1:

Dear  Sir / Madam,

Thank you very much for reviewing our manuscript and we are grateful for your comments.  Below we  answer point-by-point to your concerns:

1) You wrote:

In the introduction section, please discuss the numbers and statistics of breast cancer patients worldwide.

We are answering:

In the revised version, Introduction now contains the newly added information on the incidence and mortality of/from breast cancer in 2020 worldwide (marked by yellow in the first paragraph of Introduction in page 1). Accordingly, ref [2] has been changed (marked by yellow in the List of refs in page 35 of the revised version). We think that this is enough and we would not like to overload our paper by epidemiological data.         

2) You wrote:

In section 2, please add some discussion about oligomerization of Hsp70 and cite : Takakuwa et al., Oligomerization of Hsp70: Current perspectives on regulation and function.

We are answering:

The required discussion and respective reference [15] have been added in section 2 of the revised version (marked by yellow in pages 4 and 35).   

3) You wrote:

Figure 2 is crude and can be improved using cartoon depiction of the pathways that authors want to draw attention to.  

We are answering:

The revised version now contains three new Figures (see Figs 2-4) that demonstrate the relevant molecular pathways we would like to draw readers’ attention to.

4) You wrote:

 In line 915 "The main difficulty in creating an antitumor drug on the basis of inhibitors of HSP70 .... organs.", authors should discuss alternative strategies such as targeting co-chaperones in other cancer types, as discussed in Moses et al. Targeting the Hsp40/Hsp70 Chaperone Axis as a Novel Strategy to Treat Castration-Resistant Prostate Cancer and Nitika et al., Chemogenomic screening identifies the Hsp70 co-chaperone DNAJA1 as a hub for anticancer drug resistance.

We are answering:

The required discussion and respective references [190] and [191] have been added in the revised version (marked by yellow in pages 23 and 42).    

We hope that you’ll find the revised version improved according to your comments. Thank you in advance for your consideration.

With best wishes,

The authors: Alexander Kabakov and Vladimir Gabai

Reviewer 2 Report

This manuscript provides extensive details on how Hsp70 is involved in the pathogenesis of breast cancer. The authors describe the details related to Hsp70-mediated tumor growth and the failure of host immunity to tumor cells. Then they discuss the role of extracllular Hsp70 on breast cancer biology. The authors also include information related to different isoforms of Hsp70, followed by the potential Hsp70-targeted approaches for cancer therapy. Overall, this paper provides updates on this topic. However, the information is poorly organized, and they discuss similar information in different sections.

Author Response

Authors’ responses to Reviewer 2:

Dear  Sir / Madam,

Many thanks for reviewing our manuscript. We are grateful for your criticism with which we are mostly agreed. Below we answer to your remarks and describe what we have corrected in the revised version:

You wrote:  

This manuscript provides extensive details on how Hsp70 is involved in the pathogenesis of breast cancer. The authors describe the details related to Hsp70-mediated tumor growth and the failure of host immunity to tumor cells. Then they discuss the role of extracellular Hsp70 on breast cancer biology. The authors also include information related to different isoforms of Hsp70, followed by the potential Hsp70-targeted approaches for cancer therapy. Overall, this paper provides updates on this topic. However, the information is poorly organized, and they discuss similar information in different sections.

We are answering:

We are agreed that in the previous version, some parts of sections 5 and 6 looked as repeats of data described in sections 3 and 4. We think that some repeats may be in large review articles that consider both molecular mechanisms of a pathology and approaches to molecular targeting it. Such repeats are not bad for readers who simultaneously get a knowledge on both the molecular nature of a disease and potential ways of treating this disease with various (molecular) agents. Indeed, in the previous version of our manuscript, we sometimes mentioned certain gene-silencing vectors and/or selective inhibitors being used as tools to prove the involvement of either HSPA member in the development/pathogenesis of breast cancer (sections 3 and 4), while the same vectors or inhibitors were then described and discussed as potentially curative agents/tools to target breast cancer (sections 5 and 6). In response to your criticism, we have now tried to maximally reduce such repeating in those sections. In the revised version, each case of double mentioning of any approach or agent is now accompanied by a note addressing a reader to another section where the same approach/agent is described as well. Herein, all surplus details have been deleted. Additionally, we have made such major corrections in the revised version:

1). In subsection 5.1., listing of signs of the breast cancer pathogenesis has been totally removed from the first paragraph. The next paragraph has been substantially shortened to avoid of excessive repeating;  

2). In subsection 5.2.1., the second paragraph describing HSPA5 and USP22 has been totally removed as a surplus repeat;

3). In subsection 5.3., the first paragraph has been reduced to one phrase; the second paragraph has been totally removed; the third and fourth paragraphs have been substantially shortened;

4). In subsection 6.2.1., the second paragraph has been shortened.  

Moreover, it seems to us that the newly added four Figures (Figs. 2-4 and 6 in the revised version) also improve the organization of our manuscript and make it more readable.   

We hope that you’ll find the revised version as improved according to your comments.  

Thank you in advance for your consideration.

With best wishes,

The authors: Alexander Kabakov and Vladimir Gabai

Reviewer 3 Report

In this comprehensive review, Kabakov and Gabai detail the involvement of various cellular Hsp70 proteins in carcinogenesis – particularly breast cancer – and approaches to targeting Hsp70 as a therapeutic strategy for breast cancer. They describe cellular pathways and potential mechanisms by which this essential molecular chaperone can promote oncogenic transformation and specific strategies to counteract these pathways. Overall, it is a sweeping review of the field and could serve as a good reference point for researchers in this area.

I have a few general suggestions to improve the review and enhance its readability and accessibility:

  1. The use of figures: There are only 2 figures in the manuscript; both are very rudimentary and do not add much value to the manuscript. It would be much more useful if the authors included cartoon or visuals of the pathways which they talk about targeting and/or the specific therapeutic strategies (at least for some cases, if not all). On the other hand, the tables included currently in the manuscript are very useful.
  2. Adding a therapeutic strategies summary: As already mentioned previously, the review is very detailed and comprehensive. However for a reader, the navigability and utility will be further improved if there is a bulleted summary section for each therapeutic strategy or a final table for each strategy including the pros and cons of each approach.
  3. In the introduction section, the authors describe breast cancer subtypes classified on the basis of molecular markers expressed by tumor cells. This characterization is a very important aspect as it is related to the prognosis. It would be particularly useful if the authors included a section on the proteomic/transcriptomic profiling of each of these subtypes, focusing on the specific changes in the Hsp70 pathway proteins and their targets. This is especially relevant because the choice of the therapeutic strategy would also depend heavily on the subtype.
  4. The abstract is very generalized and could be potentially improved to better reflect the content of the review. The language style of the abstract is also distinctly different from the rest of the paper.

Author Response

Authors’ responses to Reviewer 3:

Dear  Sir / Madam,

Thank you very much for reviewing our manuscript. We are grateful for your helpful advices. Below we answer point-by-point to your comments:

You wrote:

The use of figures: There are only 2 figures in the manuscript; both are very rudimentary and do not add much value to the manuscript. It would be much more useful if the authors included cartoon or visuals of the pathways which they talk about targeting and/or the specific therapeutic strategies (at least for some cases, if not all). On the other hand, the tables included currently in the manuscript are very useful.

We are answering:

Yes, we are agreed with your remark on Figures. Now the revised version contains six Figures in total. The Figs. 2-4 and 6 are the newly added ones and these additional Figures demonstrate the relevant pathways (Figs. 2-4) and therapeutic strategies suggested (Fig. 6). Besides, the revised version contains six Tables; it seems to us that such amount of Tables is quite sufficient.  

You wrote:

Adding a therapeutic strategies summary: As already mentioned previously, the review is very detailed and comprehensive. However for a reader, the navigability and utility will be further improved if there is a bulleted summary section for each therapeutic strategy or a final table for each strategy including the pros and cons of each approach.

We are answering

In the revised version, we have added the short ‘summation’ paragraphs in the end of each (sub)section on different therapeutic strategies (marked by green in sections 5 and 6). Besides, the new Figure 6 helps to see all the major strategies (approaches) in one picture. As all the strategies are mainly at the stage of experimental models and demonstrating ‘proof-of-methods’, it is early to speculate about their pros and cons. 

You wrote:

In the introduction section, the authors describe breast cancer subtypes classified on the basis of molecular markers expressed by tumor cells. This characterization is a very important aspect as it is related to the prognosis. It would be particularly useful if the authors included a section on the proteomic/transcriptomic profiling of each of these subtypes, focusing on the specific changes in the Hsp70 pathway proteins and their targets. This is especially relevant because the choice of the therapeutic strategy would also depend heavily on the subtype.

We are answering:

Thanks for the good idea that may be a topic for a future review. However, the comparison of breast cancer subtypes on proteomic/transcriptomic profiling with focusing on HSP70 pathway/target patterns was out of topics of our review. It seems that such addition would be too much for the Introduction section and many additional pages (with the data on proteomic/transcriptomic profiling) would overload this section. Instead, we have added Table 1 where we provide data on proteomic/transcriptomic analyses showing the expression of HSPA subfamily members in different subtypes of breast cancer (see Introduction in the revised version and two respective refs [10,11] in the List of refs, all is marked by green). Although this is not exactly what you advised, we hope that this information will be helpful for readers.

You wrote:

The abstract is very generalized and could be potentially improved to better reflect the content of the review. The language style of the abstract is also distinctly different from the rest of the paper.

We are answering:

Thanks for your criticism. In response to your remark, we have corrected the abstract. The revised variant now contains more details and better reflect the content of our review (at least, we think so).  Likewise, we have tried to improve the language style in the abstract.    

It seems to us that we have satisfied the greater part of your requirements and you’ll find the revised version as the improved one. Thank you in advance for your consideration.

With best wishes,

The authors: Alexander Kabakov and Vladimir Gabai

Round 2

Reviewer 2 Report

The manuscript has been improved.

Reviewer 3 Report

In the revised version, the authors have taken majority of the suggestions and critique into account and the manuscript now looks significantly improved.